# Multimodal HLA-I genotype regulation by human cytomegalovirus US10 and resulting surface patterning

Carolin Gerke[1,2,3,4], Liane Bauersfeld[1,2], Ivo Schirmeister[1,2], Chiara Noemi-Marie Mireisz[5], Valerie Oberhardt[2,6], Lea Mery[1,2], Di Wu[1,2], Christopher Sebastian Jürges[7], Robbert M Spaapen[8,9], Claudio Mussolino[2,10,11], Vu Thuy Khanh Le-Trilling[12], Mirko Trilling[12,13], Lars Dölken[7,14], Wolfgang Paster[15], Florian Erhard[7], Maike Hofmann[2,6], Andreas Schlosser[5], Hartmut Hengel[2], Frank Momburg[16], Anne Halenius[1,2]*

[1]Institute of Virology, Medical Center University of Freiburg, Freiburg, Germany; [2]Faculty of Medicine, University of Freiburg, Freiburg, Germany; [3]Spemann Graduate School of Biology and Medicine (SGBM), University of Freiburg, Freiburg, Germany; [4]Faculty of Biology, University of Freiburg, Freiburg, Germany; [5]Rudolf Virchow Center, Center for Integrative and Translational Bioimaging, University of Würzburg, Würzburg, Germany; [6]Department of Medicine II (Gastroenterology, Hepatology, Endocrinology and Infectious Diseases), Medical Center University of Freiburg, Freiburg, Germany; [7]Institute for Virology and Immunobiology, University of Würzburg, Würzburg, Germany; [8]Department of Immunopathology, Sanquin Research, Amsterdam, Netherlands; [9]Landsteiner Laboratory, Amsterdam UMC, University of Amsterdam, Amsterdam, Netherlands; [10]Institute for Transfusion Medicine and Gene Therapy, Medical Center University of Freiburg, Freiburg, Germany; [11]Center for Chronic Immunodeficiency, Medical Center University of Freiburg, Freiburg, Germany; [12]Institute for Virology, University Hospital Essen, University of Duisburg-Essen, Essen, Germany; [13]Institute for the Research on HIV and AIDS-associated Diseases, University Hospital Essen, Essen, Germany; [14]Institute of Virology, Hannover Medical School, Hannover, Germany; [15]St. Anna Children's Cancer Research Institute (CCRI), Vienna, Austria; [16]Clinical Cooperation Unit Applied Tumor Immunity, German Cancer Research Center, National Center for Tumor Diseases (NCT), Heidelberg University Hospital, Heidelberg, Germany

*For correspondence:
anne.halenius@uniklinik-freiburg.de

Competing interest: The authors declare that no competing interests exist.

**Abstract** Human leucocyte antigen class I (HLA-I) molecules play a central role for both NK and T-cell responses that prevent serious human cytomegalovirus (HCMV) disease. To create opportunities for viral spread, several HCMV-encoded immunoevasins employ diverse strategies to target HLA-I. Among these, the glycoprotein US10 is so far insufficiently studied. While it was reported that US10 interferes with HLA-G expression, its ability to manipulate classical HLA-I antigen presentation remains unknown. In this study, we demonstrate that US10 recognizes and binds to all HLA-I (HLA-A, -B, -C, -E, -G) heavy chains. Additionally, impaired recruitment of HLA-I to the peptide loading complex was observed. Notably, the associated effects varied significantly depending on HLA-I genotype and allotype: (i) HLA-A molecules evaded downregulation by US10, (ii) tapasin-dependent HLA-B molecules showed impaired maturation and cell surface expression, and (iii) $\beta_2$m-assembled HLA-C, in particular HLA-C*05:01 and -C*12:03, and HLA-G were strongly retained in complex with US10 in the endoplasmic reticulum. These genotype-specific effects on HLA-I were confirmed through unbiased HLA-I ligandome analyses. Furthermore, in HCMV-infected fibroblasts inhibition

of overlapping US10 and US11 transcription had little effect on HLA-A, but induced HLA-B antigen presentation. Thus, the US10-mediated impact on HLA-I results in multiple geno- and allotypic effects in a so far unparalleled and multimodal manner.

## Editor's evaluation

HLA class I (HLA-I) molecules play a central role for Natural killer- as well as T-cell-mediated responses crucial to control the herpesvirus human cytomegalovirus (HCMV), and there are multiple viral immune evasins that target the protective function of HLA-I. This valuable study reports on the HCMV immune evasin US10 and shows that it recognizes and binds to all HLA-I (HLA-A,-B,-C,-E,-G) heavy chains and affects their function in different ways. These convincing findings reveal novel functions of the viral glycoprotein US10. The study improves our understanding of this complex and potentially life-threatening herpesvirus and its interaction with our immune system.

## Introduction

The human cytomegalovirus (HCMV) belongs to the β-herpesviruses and establishes a life-long persistent infection in humans alternating between phases of latency and reactivation. Although clinical manifestations are mainly observed in immunocompromised patients, HCMV also affects the immune system of healthy individuals (*Brodin et al., 2015*). For example, an expansion of memory-like NK cells as well as of CD8[+] memory T cells are frequent observations in HCMV-positive individuals (*Lübke et al., 2020*; *Rölle and Brodin, 2016*; *Waller et al., 2008*). These cytotoxic immune effector cells are crucial for HCMV control (*Sylwester et al., 2005*; *Venema et al., 1994*). While specific antigenic peptide ligands presented by major histocompatibility complex class I (MHC-I) molecules on infected cells activate CD8[+] T-cells, NK cells express various inhibiting and activating receptors that recognize MHC-I both in peptide-independent and -dependent manners.

In humans, MHC-I molecules are encoded by three classical (A, B, C) and three non-classical (E, F, G) human leucocyte antigen class I (HLA-I) loci located on chromosome 6 (*Koller et al., 1989*). Classical HLA-I are characterized by a high degree of polymorphism, whereas non-classical HLA-I show low levels of heterogeneity. The MHC-I maturation process begins with a co-translational translocation of the heavy chain (HC) into the endoplasmic reticulum (ER), where it folds and assembles with β$_2$-microglobulin (β$_2$m). To acquire a peptide ligand, MHC-I is assisted by the chaperones tapasin, ERp57, and calreticulin, as well as the transporter associated with antigen processing (TAP), together forming the peptide loading complex (PLC) (*Blees et al., 2017*; *Hulpke and Tampé, 2013*). While TAP transports cytosolic peptides generated by the proteasome into the ER (*Michalek et al., 1993*), tapasin serves as an adapter chaperone linking TAP and HLA-I. Additionally, tapasin fulfills a crucial role in peptide editing and optimization (*Wearsch and Cresswell, 2007*; *Williams et al., 2002*). Peptide-loaded MHC-I is transported to the cell surface via the secretory pathway enabling immune cell surveillance of MHC-I expression and antigen presentation.

Different from most viruses, cytomegaloviruses encode several proteins targeting MHC-I molecules. HCMV applies various strategies (*Halenius et al., 2015*): US2 and US11 initiate degradation of the MHC-I HC (*Jones and Sun, 1997*; *Wiertz et al., 1996*), US3 strongly retains MHC-I in the ER and blocks tapasin function (*Jones et al., 1996*; *Park et al., 2004*), and US6 blocks the peptide transport by TAP (*Ahn et al., 1997*; *Hengel et al., 1997*; *Lehner et al., 1997*). These immunoevasins belong to the *US2* and *US6* gene families, also including US9, which targets the MHC-I-like molecule MICA*008 with its signal peptide (*Seidel et al., 2021*), and US7 and US8, which antagonize Toll-like receptor signaling (*Park et al., 2019*). Like all members of these families, US10 is a type I transmembrane glycoprotein and localized in the ER (*Huber et al., 2002*). Immunoprecipitation experiments demonstrated binding to MHC-I as well as a delay of MHC-I maturation in the presence of US10 (*Furman et al., 2002*). Furthermore, US10 was shown to be involved in the degradation of the non-classical MHC-I molecule HLA-G (*Park et al., 2010*). Altogether, a role of US10 as an MHC-I immunoevasin is clearly indicated, but it is incompletely understood to which extent US10 binding to HLA-I results in impaired HLA-I function.

Our previous studies revealed pronounced genotype-specific differences in HLA-I regulation by US11: HLA-A is strongly targeted by US11-mediated proteasomal degradation, whereas HLA-B can

**eLife digest** During a viral infection, the immune system must discriminate between healthy and infected cells to selectively kill infected cells. Healthy cells have different types of molecules known collectively as HLA-I on their surface. These molecules present small fragments of proteins from the cell, called antigens, to patrolling immune cells, known as CTLs or natural killer cells.

While CTLs ignore antigens from human proteins (which indicate the cell is healthy), they can bind to and recognize antigens from viral proteins, which triggers them to activate immune responses that kill the infected cell. However, some viruses can prevent infected cells from presenting HLA-I molecules on their surfaces as a strategy to evade the immune system. Natural killer cells have evolved to overcome this challenge. They bind to the HLA-I molecules themselves, which causes them to remain inactive. However, if the HLA-I molecules are missing, the NK cells can more easily switch on and kill the target cell.

The human cytomegalovirus is a common virus that causes lifelong infection in humans. Although it rarely causes illness in healthy individuals, it can be life-threatening to newborn babies and for individuals with weakened immune systems. One human cytomegalovirus protein known as US10 was previously found to bind to HLA-I without reducing the levels of these molecules on the surface of the cell. However, its precise role remained unclear.

Gerke et al. used several biochemical and cell biology approaches to investigate whether US10 manipulates the quality of the three types of HLA-I, which could impact both CTL and NK cell recognition. The experiments showed that US10 acted differently on the various kinds of HLA-I. To one type, it bound strongly within the cell and prevented it from reaching the surface. US10 also prevented another type of HLA-I from maturing properly and presenting antigens but did not affect the third type of HLA-I.

These findings suggest that US10 interferes with the ability of different HLA-I types to present antigens in specific ways. Further research is needed to measure how US10 activity affects immune cells, which may ultimately aid the development of new therapies against human cytomegalovirus and other similar viruses.

escape proteasomal targeting (*Zimmermann et al., 2019*; *Zimmermann et al., 2024*). Thus, we hypothesized that also US10 possesses HLA-I genotypic, or even allotypic preferences, and aimed at elucidating the targeting spectrum of US10 to gain insights into its role in control of HLA-I functions.

Here, we describe a pronounced US10 selectivity for HLA-I targets highlighting an intricate role for US10 as an HLA-I immunoevasin.

## Results

### Geno- and allotype-specific downregulation of HLA-I cell surface expression by US10

To gain deeper insight into the immunoevasive role of US10, we set out to determine its HLA-I preferences using a quantitative flow cytometry assay (the general principle of the assay was described in *Zimmermann et al., 2019*). A panel of plasmids encoding N-terminally HA-tagged HLA-I molecules, or CD99 as control, were co-transfected with an EGFP-expressing vector encoding US10 or an HCMV control protein (RL8). HLA-I surface expression was determined by anti-HA staining of EGFP-positive cells at 20 hr post-transfection. The HLA-I-related molecules MICA*004 and MR1, and the murine H-2K$^b$ were not downregulated by US10, while US10 displayed a wide-ranging impact on HLA-I regulation (*Figure 1A and B*). HLA-G surface expression was almost entirely absent on US10-expressing cells. Also, the expression of other HLA-I molecules, namely HLA-B*44:02, -C*05:01, and -E, was decreased, while HLA-C*04:01, -C*12:03, -B*08:01, and -B*15:03 were affected to a lesser degree. Remarkably, all five HLA-A allotypes were resistant to US10-mediated downregulation.

We next analyzed US10 binding to HLA-I. To this end, US10 and HA-tagged HLA-I were transiently expressed in HeLa cells. The cells were metabolically labeled for 2 hr, and using an anti-HA antibody a co-immunoprecipitation experiment was performed (*Figure 1C*). HA-CD99 and untagged HLA-B*44:02 were included as negative controls. US10 bound to all tested HLA-I molecules (allotypes

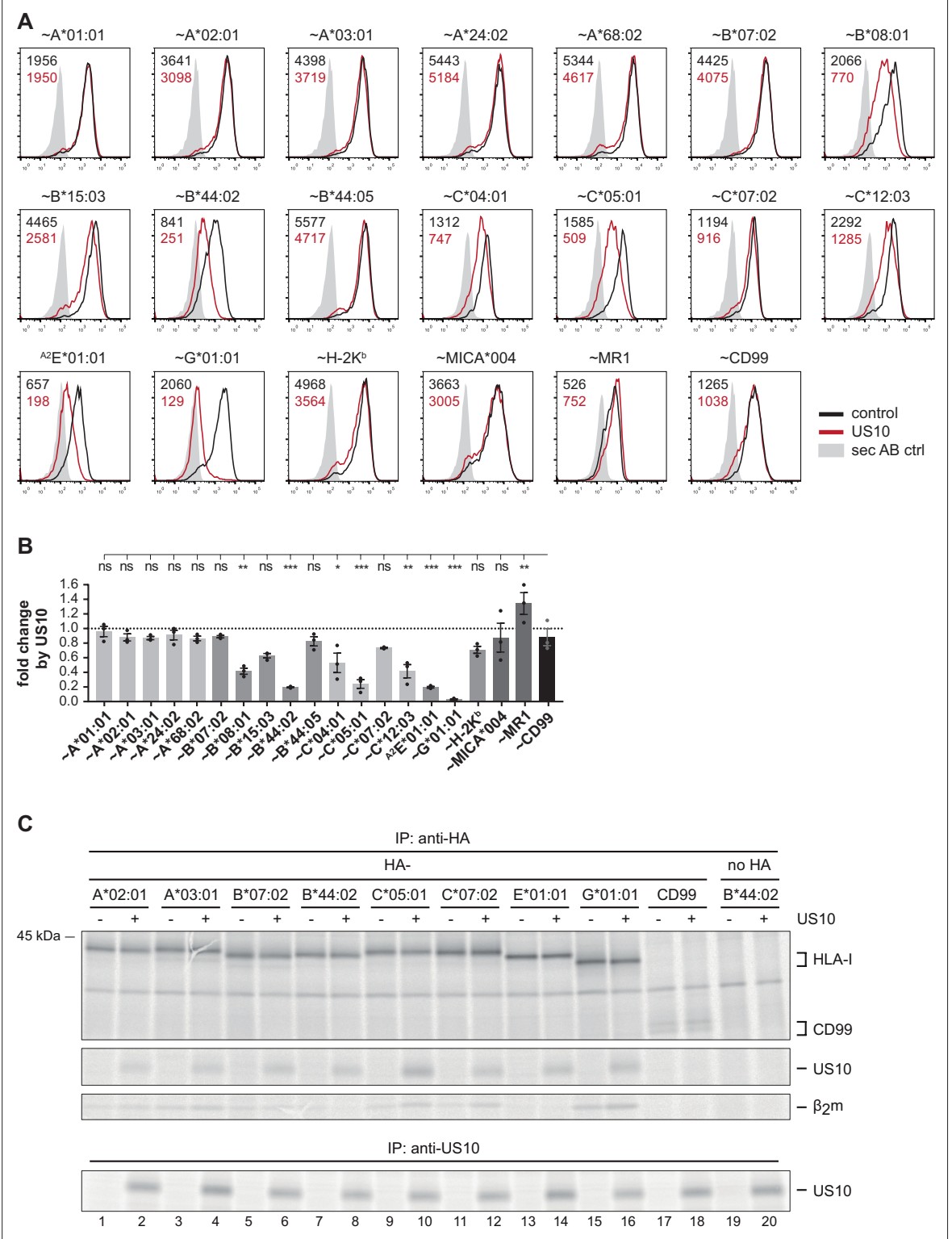

**Figure 1.** Geno- and allotype-specific regulation of HLA-I by US10. (**A**) HeLa cells were transiently co-transfected with plasmids encoding HA-tagged (~) molecules or non-tagged HLA-$^{A2}$E*01:01 (HLA-E was expressed with an HLA-A*02 signal peptide, a natural HLA-E ligand) and plasmids with either US10 or a control protein together with an IRES-*EGFP* cassette. Surface expression was determined by flow cytometry (anti-HA or anti-HLA-E) on EGFP-positive cells. Representative histograms are shown. (**B**) Fold change of surface expression by US10 was calculated as the ratio of the median fluorescence intensity (MFI) of US10-expressing cells compared to control transfected cells. Dots represent individual values and bars mean values ± SEM from three independent experiments (biological replicates). Significance compared to the HA-CD99 control was calculated using one-way

*Figure 1 continued on next page*

*Figure 1 continued*

ANOVA followed by Dunnett's multiple comparison test. (**C**) HeLa cells were transiently transfected as described in (**A**) and metabolically labeled for 2 hr. Digitonin cell lysates were prepared, and immunoprecipitations using anti-HA or anti-US10 were performed and separated by SDS-PAGE with subsequent detection by autoradiography (*Figure 1—source data 1*). One of two independent experiments is shown.

The online version of this article includes the following source data for figure 1:

**Source data 1.** Immunoprecipitation in *Figure 1C*.

of HLA-A, -B, -C, and HLA-E and -G), and most strongly to HLA-C*05:01. Thus, US10's ability to bind HLA-I does not explain the different expression patterns on the cell surface. Furthermore, in contrast to previous observations (*Park et al., 2010*), we did not observe destabilization of HLA-G or other HLA-I molecules in US10-expressing cells (*Figure 1C*).

## US10 blocks HLA-I interaction with the PLC

Prior studies demonstrated that US10 retains HLA-I in the ER (*Furman et al., 2002*). However, it was not determined whether HLA-I allomorph-dependent differences in retention exist. Therefore, we assessed maturation of endogenous HLA-I (HLA-A*68:02, -B*15:03, and -C*12:03) in wild-type and US10-expressing HeLa cells. When separated by SDS-PAGE, HLA-A*68:02 can be distinguished from the two other HLA-I allomorphs due to faster migration (*Zimmermann et al., 2019*). We found that HLA-A*68:02 was clearly less retained by US10 than HLA-B*15:03/C*12:03, as judged from the acquisition of endoglycosidase H (EndoH)-resistant glycans in a pulse-chase experiment (*Figure 2A and B*).

A rate determining factor for the ER exit of HLA-I molecules is the average time required to load a stabilizing ligand in the peptide binding groove. The dependency on the PLC for efficient peptide loading varies greatly between HLA-I molecules (*Bashirova et al., 2020*). To assess whether the PLC is involved in the US10-mediated delay of HLA-I maturation, HLA-I interaction with the PLC was analyzed (*Figure 2C*). Co-immunoprecipitation using antibodies directed either against ERp57 or tapasin revealed that the overall level of PLC-associated HLA-I was greatly reduced in cells expressing US10, despite the increase in EndoH-sensitive HLA-I in these cells (*Figure 2C*, lanes 9–12). This indicates that larger quantities of HLA-I molecules remained in the ER, but these were not able to interact with the PLC. Interestingly, we observed that HLA-A*68:02 was less affected than HLA-B*15:03/-C*12:03; the percentage of HLA-A*68:02 in the PLC compared to the total amount of HLA-I in the PLC was increased in US10-expressing cells, whereas it was decreased for HLA-B*15:03/-C*12:03 (*Figure 2D*). Thus, HLA-A*68:02 access to the PLC was less disturbed by US10 than observed for HLA-B*15:03/-C*12:03. A co-immunoprecipitation of US10 with the PLC was not observed in this experiment (*Figure 2—figure supplement 1A*). Either US10 is not interacting with the PLC or the PLC might be saturated with unlabeled US10.

To investigate whether US10 can interact with the PLC at all, we transiently transfected US10 and tapasin and analyzed the interaction with ERp57 using near steady-state conditions (metabolic labeling for 3 hr; *Figure 2E*). While the endogenous level of tapasin barely allowed detection of US10 co-immunoprecipitation (*Figure 2E*, lane 2), overexpression of tapasin increased the level of US10 co-immunoprecipitation (lane 3). In cells lacking HLA-I expression (HLA-I was knocked out using a lentiviral CRISPR/Cas9 system; *Figure 2—figure supplement 1B and C*), the interaction with US10 remained discernible, albeit very faintly (lane 4). Moreover, in samples with tapasin overexpression anti-US10 rabbit antiserum co-immunoprecipitated tapasin. This was the case also in the absence of HLA-I expression, but with decreased intensity. HLA-I and ERp57 were not detected under these conditions (lanes 7–8). Taken together, these findings suggest that US10 may interact directly with tapasin, and possibly, this interaction blocks HLA-I binding to the PLC.

## Higher tapasin dependency in HLA-B correlates with increased sensitivity to US10

The pronounced sensitivity of HLA-B*44:02 and -B*08:01 to US10, in contrast to the lack of sensitivity observed for HLA-B*44:05 and -B*07:02, and the observation that US10 blocks HLA-I-interaction with the PLC, prompted us to investigate whether the degree of HLA-I tapasin-dependency correlated with US10 sensitivity. We engineered tapasin knockout HeLa cells (*Figure 3—figure supplement 1A*) and compared the effect of tapasin deficiency with the effect of ectopic US10 expression (*Figure 3A*,

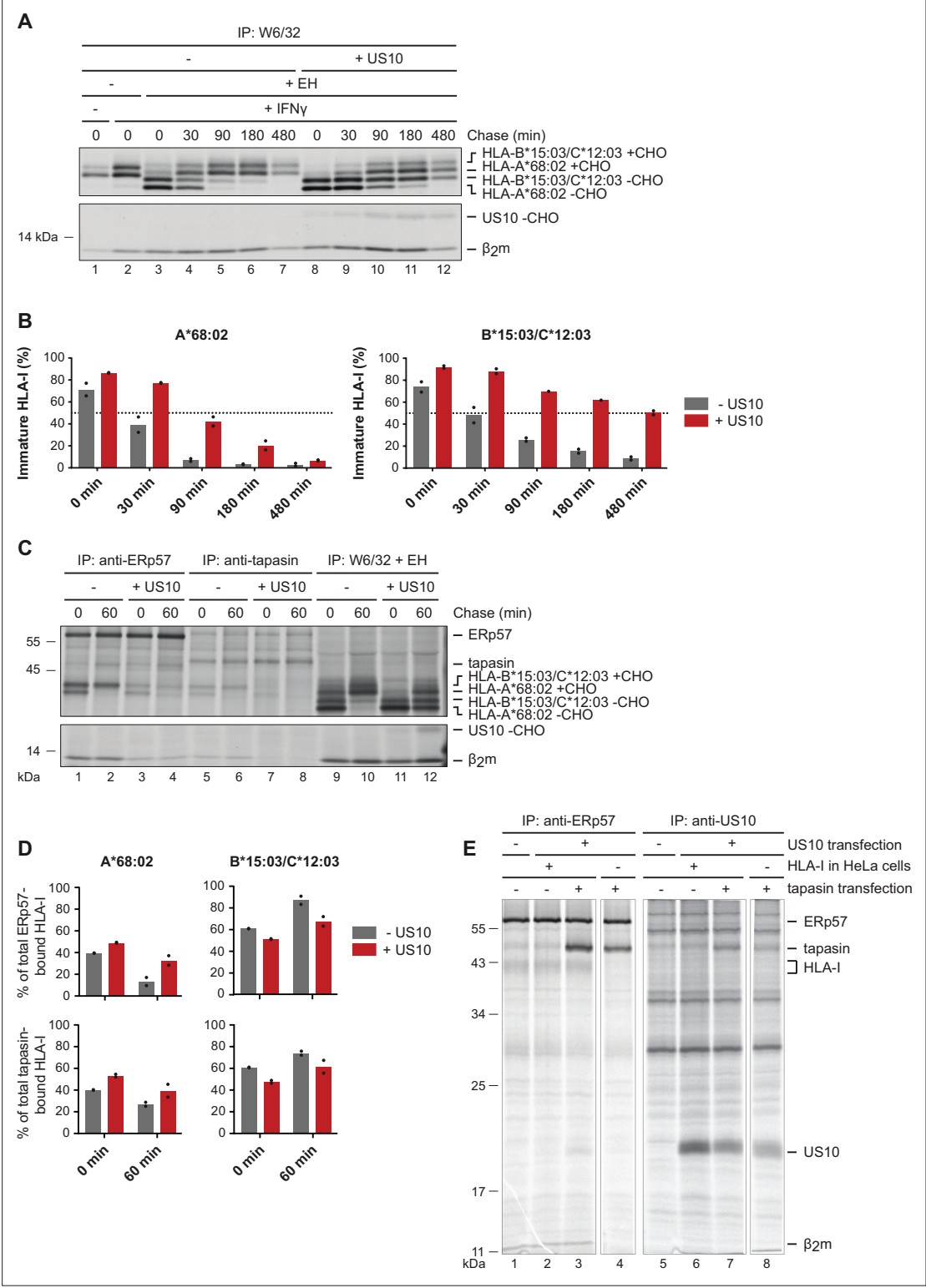

**Figure 2.** US10 blocks human leucocyte antigen class I (HLA-I) interaction with the peptide loading complex (PLC). (**A**) Control HeLa cells or cells stably expressing US10 were induced by IFNγ overnight and subsequently metabolically labeled for 30 min and chased as indicated. After immunoprecipitation by W6/32, proteins were digested by EndoH (-CHO, deglycosylated proteins; +CHO, resistant glycans) as indicated and separated by SDS-PAGE. Labeled proteins were detected by autoradiography (*Figure 2—source data 1*). (**B**) The intensities of single HLA-I heavy chain (HC) bands in (**A**) were quantified, and the percentage of immature molecules compared to the total amount (sum of immature and mature) was calculated and depicted from two independent experiments (biological replicates). (**C**) Immunoprecipitation from HeLa cells or cells stably expressing US10 was

*Figure 2 continued on next page*

*Figure 2 continued*

performed as in (**A**) but with modified chase times and without IFNγ treatment. Antibodies applied for immunoprecipitations are indicated (*Figure 2—source data 2*). (**D**) Band intensities of HLA-I HCs in anti-ERp57 and anti-tapasin immunoprecipitations from (**C**) were quantified and the amount of the HLA-A*68:02 HC (left panel) and HLA-B*15:03/-C*12:03 HC (right panel) was calculated as the percentage of total PLC-bound HLA-I (sum of both HC bands). Dots represent individual values from two independent experiments (biological replicates). (**E**) Wild-type or HLA-I KO HeLa cells were transiently transfected with US10 and tapasin-expressing plasmids as indicated. At 20 hr post-transfection, cells were metabolically labeled for 3 hr. Immunoprecipitation was performed with anti-ERp57 or anti-US10 antibodies (*Figure 2—source data 3*). One of two independent experiments is shown in panels (**A**), (**C**), and (**E**).

The online version of this article includes the following source data and figure supplement(s) for figure 2:

**Source data 1.** Immunoprecipitation in *Figure 2A*.

**Source data 2.** Immunoprecipitation in *Figure 2C*.

**Source data 3.** Immunoprecipitation in *Figure 2E*.

**Figure supplement 1.** US10 blocks human leucocyte antigen class I (HLA-I) interaction with the peptide loading complex (PLC).

**Figure supplement 1—source data 1.** Immunoprecipitation in *Figure 2—figure supplement 1A*.

*Figure 3—figure supplement 1B*). In this analysis, HLA-I surface expression was enhanced by treating the cells with IFNγ because of the otherwise low HLA-I expression in the absence of tapasin. The level of HLA-I tapasin-dependency obtained from our analysis correlated well with data from others (*Bashirova et al., 2020*), and IFNγ treatment did not change the pattern of HLA-I regulation by US10. Since HLA-A allotypes were resistant to US10, no correlation between tapasin dependency and US10 sensitivity was observed for HLA-A (*Figure 3B*). In contrast, HLA-B tapasin dependency correlated strongly with US10-mediated downregulation. Curiously, even though HLA-C molecules showed different degrees of regulation by US10, this did not correlate with the tapasin dependency of the molecules.

In addition, the analysis of HLA-I expression in tapasin knockout cells showed that US10 was able to further reduce surface expression of US10-sensitive HLA-I such as HLA-B*15:03, -C*05:01, and -G*01:01 (*Figure 3C*), indicating that US10 can promote HLA-I downregulation in a tapasin-independent manner. Taken together, the mode of US10 targeting of HLA-I is foremost determined by their genotypes and not by tapasin function.

## US10 binding to β₂m/HC heterodimers correlates with HLA-I ER retention

The anti-HA antibody used for co-immunoprecipitation in *Figure 1C* does not reveal which conformation of HLA-I is preferentially bound by US10. To address this issue, we used HeLa cells lacking HLA-I expression and transiently co-expressed US10 or a control protein together with HA-tagged HLA-I (with a mutated gRNA target site). After metabolic labeling of cells, the mAb W6/32 was applied to assess binding of US10 to β₂m-assembled HLA-I (*Figure 4A*). The binding strength (signal intensity) of US10 was compared to the maximum intensity of US10 (HLA-G-bound US10). Remarkably, a pronounced difference in US10 binding to β₂m-assembled HLA-G and HLA-C*05:01 (1.0- and 0.7-fold of max, respectively) compared to assembled HLA-A, -B, and -E allotypes (all below 0.1-fold of max; *Figure 4B*) was observed. Binding of US10 to assembled HLA-C*07:02 showed a slight increase (0.2-fold). Application of an anti-β₂m mAb confirmed the increased US10 binding to β₂m-assembled HLA-G and -C (*Figure 4—figure supplement 1A and B*). These findings suggest that the primary basis for the anti-HA-mediated co-immunoprecipitation of US10 with HLA-A, -B, and -E allotypes is an interaction with their free HCs. Furthermore, while the anti-HA immunoprecipitation showed a slight increase in all HLA-I HCs (except for HLA-B*44:02) when co-expressed with US10, this was pronounced only for HLA-C and -G after W6/32 immunoprecipitation (*Figure 4A*, *Figure 4—figure supplement 1C*). Furthermore, an EndoH digest revealed a strong retention of β₂m-assembled HLA-C*05:01 by US10, whereas HLA-A*02:01 and -B*07:02 gained EndoH-resistant glycans in the presence of US10 (*Figure 4C*). Maturation of HLA-B*44:02 was not visible in this experimental setting.

To analyze the role of β₂m for US10 binding to HLA-I HCs, we knocked out β₂m from HLA-I KO HeLa cells (*Figure 4—figure supplement 1D*). The cells were transiently transfected to express US10 and HA-tagged HLA-I with and without β₂m. Subsequently, the strength of US10 binding to HLA-I was measured using an anti-HA antibody. The highest level of US10 was bound to HLA-C*05:01 and expression of β₂m did not markedly change this (*Figure 4D and E*). In contrast, upon co-expression of

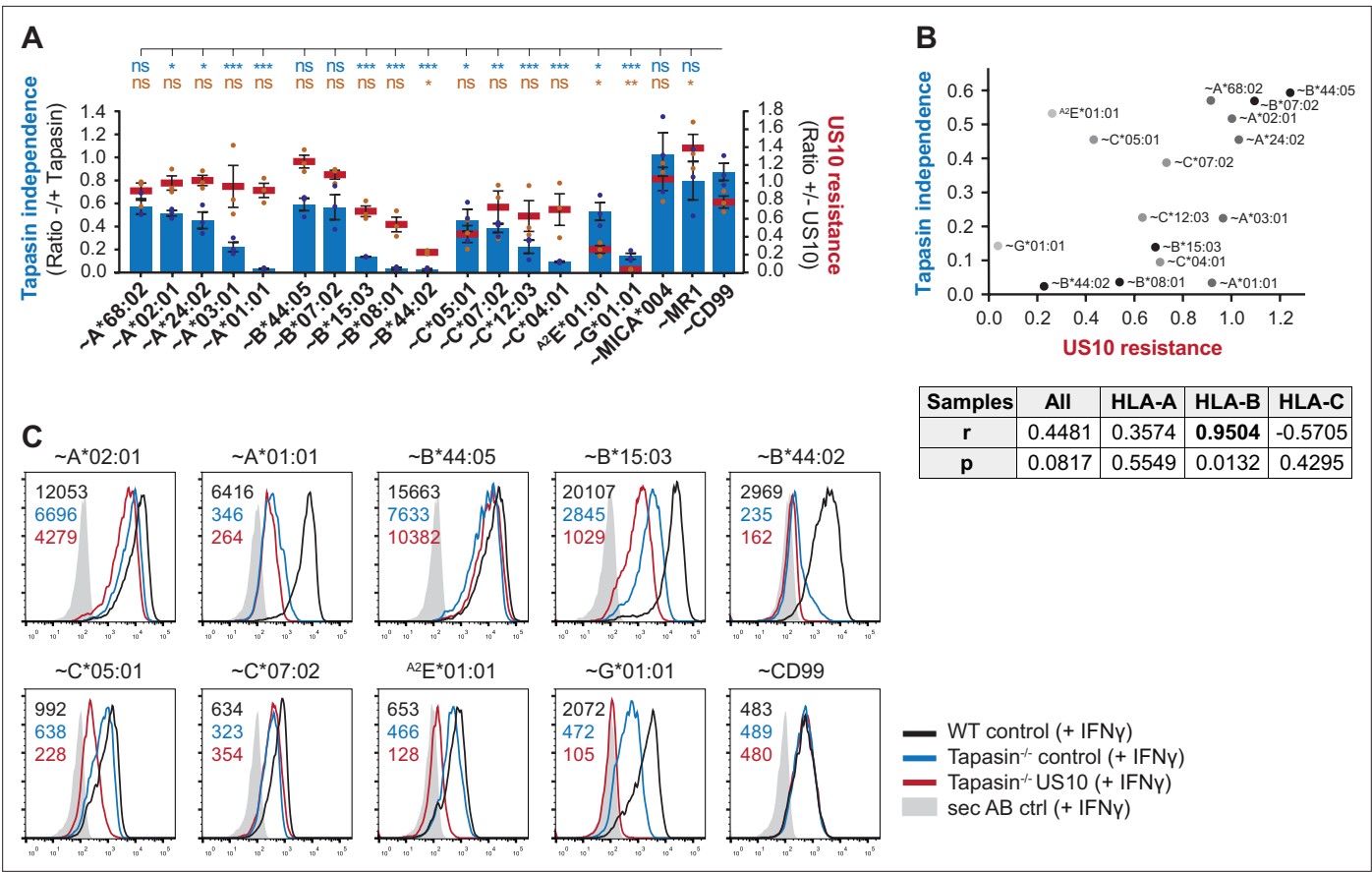

**Figure 3.** Higher tapasin dependency in HLA-B correlates with increased sensitivity to US10. (**A**) Wild-type or tapasin knockout HeLa cells were transiently co-transfected as indicated and treated with IFNγ overnight. Cell surface expression of the HA-tagged molecules or non-tagged HLA-A2E*01:01 was determined (representative histograms in *Figure 3—figure supplement 1B*) as in *Figure 1A*. US10 resistance was calculated as the ratio of the median fluorescence intensity (MFI) of US10-expressing cells compared to control cells (red lines). Tapasin independence was calculated as the ratio of the MFI of tapasin knockout cells compared to wild-type cells (blue bars). Dots represent individual values and bars mean values ± SEM from three independent experiments (biological replicates). Significance compared to the HA-CD99 control was calculated using one-way ANOVA followed by Dunnett's multiple comparison test. (**B**) Two-tailed correlation analysis of the results from (**A**). (**C**) Flow cytometry analysis performed as in (**A**) including US10 in tapasin knockout cells. Representative histograms from one of three independent experiments (biological replicates) are shown.

The online version of this article includes the following source data and figure supplement(s) for figure 3:

**Figure supplement 1.** HLA-B regulation by US10 correlates with HLA-B tapasin dependency.

**Figure supplement 1—source data 1.** Immunoprecipitation in *Figure 3—figure supplement 1A*.

β2m US10, binding to the HLA-A and -B molecules was reduced. Loss of binding was most pronounced for HLA-A*02:01 followed by -B*07:02, and -B*44:02.

Altogether, these experiments suggest that US10 has a higher affinity for HLA-C*05:01 than for the investigated HLA-A and -B allotypes. Moreover, US10 binding to HLA-C*05:01 remains intact after β2m-assembly, accompanied by a strong retention in HLA-C*05:01 in the ER. The most likely explanation for the reduced US10 binding to HLA-A*02:01 and -B*07:02 after β2m expression is that these molecules are not retained anymore by US10, but the interaction is interrupted and the molecules leave the ER. To test this, we expressed HLA-A*02:01 and US10 in HLA-I KO cells and assessed interaction with the W6/32 mAb. When the cells were treated with brefeldin A, which blocks trafficking between the ER and Golgi compartments, more HLA-A*02:01 remained EndoH sensitive and clearly more US10 was co-immunoprecipitated (*Figure 4F*). This was also observed to some extent for HLA-B*07:02 (*Figure 4—figure supplement 1E*). Therefore, US10 does not directly compete with β2m for binding to the HLA-I HC, instead, the data suggest that the affinity of US10 for assembled HLA-A and -B proteins is reduced upon HC assembly with β2m. This change in affinity could be due to the US10-HC interaction site being partially blocked by β2m.

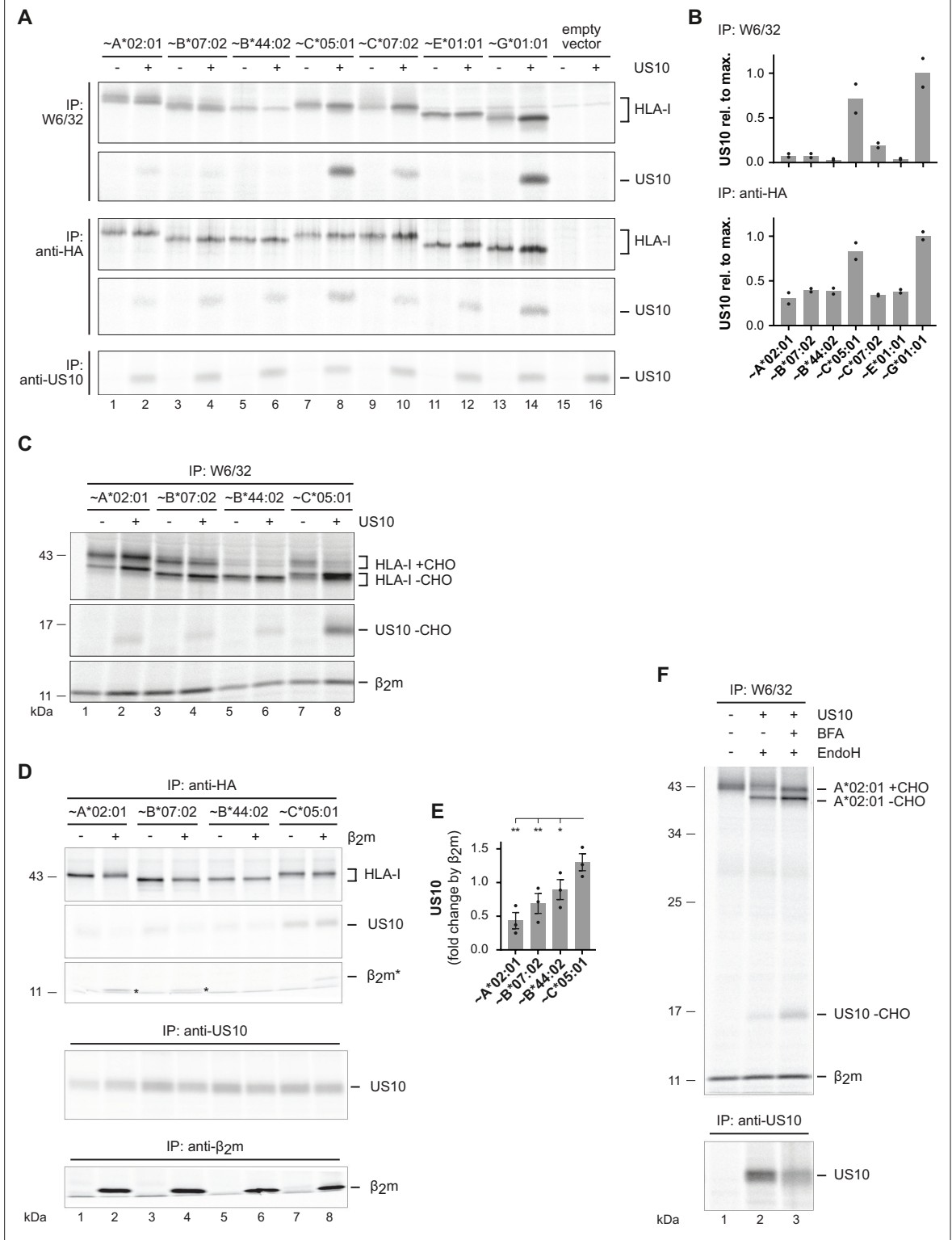

**Figure 4.** US10 binding to β₂m/HC heterodimers correlates with human leucocyte antigen class I (HLA-I) endoplasmic reticulum (ER) retention. (**A**) HLA-I KO HeLa cells were transiently co-transfected with indicated HA-HLA-I-expressing plasmids comprising a mutated gRNA binding site together with a US10- or a control-pIRES-EGFP plasmid. To improve assembly of HLA-E, UL40 (comprising an HLA-E ligand) was expressed with HLA-E. Cells were metabolically labeled for 2 hr and immunoprecipitations were performed as in *Figure 1C*; antibodies were applied as indicated on the left (*Figure 4—source data 1 and 2*). (**B**) Relative signal strengths from single bands of US10 in the W6/32 and (upper panel) anti-HA immunoprecipitation (lower panel) samples are shown. Dots represent individual values and bars mean values thereof from two independent experiments (biological replicates).

*Figure 4 continued on next page*

*Figure 4 continued*

The ratio (US10/control) of single bands of HLA-I HCs in the anti-HA and W6/32 immunoprecipitation samples is shown. Dots represent individual values and bars mean values thereof from two independent experiments (biological replicates). (**C**) HLA-I KO HeLa cells were transfected and treated as in (**A**). Immunoprecipitation was performed with W6/32 and subsequently an EndoH digest was performed (*Figure 4—source data 3*). (**D**) HLA-I/ $\beta_2$m, double KO HeLa cells were transiently transfected with US10, HA-tagged HLA-I, and $\beta_2$m as indicated. At 20 hr post-transfection, cells were metabolically labeled for 2 hr and immunoprecipitation was performed as indicated (*Figure 4—source data 4*). One of three independent experiments is shown. (**E**) The intensity of the US10 bands co-immunoprecipitated with anti-HA was quantified, and the ratios of the samples with and without $\beta_2$m were determined from three independent experiments (biological replicates). Significance was calculated using one-way paired ANOVA followed by Dunnett's multiple comparison test. (**F**) HLA-I KO HeLa cells were transfected with HA-HLA-A*02:01 and US10 or a control plasmid. At 20 hr post-transfection, cells were treated with brefeldin A (BFA) during metabolic labeling for 2 hr. Subsequently, an immunoprecipitation using anti-HA was performed. Indicated samples were subjected to EndoH digestion prior to SDS-PAGE separation (*Figure 4—source data 5*). One of two independent experiments is shown in (**A**), (**C**), and (**F**).

The online version of this article includes the following source data and figure supplement(s) for figure 4:

**Source data 1.** Immunoprecipitations with W6/32 and anti-HA in *Figure 4A*.

**Source data 2.** Immunoprecipitation with anti-US10 in *Figure 4A*.

**Source data 3.** Immunoprecipitation in *Figure 4C*.

**Source data 4.** Immunoprecipitation in *Figure 4D*.

**Source data 5.** Immunoprecipitation in *Figure 4F*.

**Figure supplement 1.** US10 forms stable complexes with assembled HLA-C and -G, but not with HLA-A and -B.

**Figure supplement 1—source data 1.** Immunoprecipitations in *Figure 4—figure supplement 1A*.

**Figure supplement 1—source data 2.** Immunoprecipitations in *Figure 4—figure supplement 1E*.

## Quantitative HLA-I ligandome analysis confirms genotype-dependent effects by US10

So far, our analysis suggested a genotype-dependent effect of US10 on HLA-I: HLA-A escapes retention, HLA-B downregulation correlates with tapasin dependency, and HLA-C is retained and stabilized by US10 in a $\beta_2$m-assembled form in the ER. To assess how these distinct phenotypes affect the HLA-I ligandome of US10-expressing cells, we generated HeLa cells with a doxycycline-inducible TagBFP-T2A-US10 construct. A cell clone was selected ('#5') that showed a similar HLA-C regulation as the main population of the parental cell pool (*Figure 5—figure supplement 1A*). After doxycycline treatment, 97% of the cells of clone #5 expressed TagBFP (*Figure 5—figure supplement 1A*). A single US10-specific band could be visualized by metabolic labeling of doxycycline-treated TagBFP-T2A-US10 cell clones, both by W6/32 co-immunoprecipitation and direct US10 immunoprecipitation, demonstrating correct expression of the fusion construct (*Figure 5—figure supplement 1B*). Without doxycycline treatment, clone #5 did not express US10 at detectable levels (*Figure 5—figure supplement 1C*). After doxycycline treatment of clone #5, the surface expression of HLA-C*12:03 was reduced to 20% and that of HLA-B*15:03 to 60% compared to untreated cells (*Figure 5A*).

Having verified US10 expression and downregulation of HLA-B*15:03 and -C*12:03 upon doxycycline treatment, a quantitative ligandomics experiment using pulsed stable isotope labeling by amino acids in cell culture (pSILAC) was implemented. To that end, we combined ca. $10^8$ cells of clone #5 that were doxycycline-treated and pulse-labeled for 24 hr with heavy amino acids with an equivalent number of untreated cells pulse-labeled with medium-heavy amino acids. Subsequently, we conducted affinity purification of HLA-I peptides and subjected them to analysis via nanoLC-MS/MS. (*Figure 5B*). In total, we identified and quantified 2909 HLA-I-presented peptides (1% false discovery rate, FDR), 1525 predicted by NetMHCpan as binder for HLA-A*68:02, 728 for HLA-B*15:03, and 656 for HLA-C*12:03. A comparison of peptide intensities verified the anticipated genotype-dependent modulation of the HLA-I ligandome (*Figure 5C*). While HLA-A*68:02 peptides were mostly unaffected by US10, HLA-B*15:03 peptides were globally decreased, and HLA-C*12:03 peptides globally increased. The observed increase in HLA-C*12:03 peptides, together with the observed reduced surface expression of HLA-C*12:03, indicated that US10 stabilized peptide-bound HLA-C*12:03 in the ER.

Through further analysis of HLA-B*15:03 ligands, it was observed that the common anchor residues at position 2 (consensus binding motif in *Figure 5D*) were reduced by US10 (*Figure 5E*). This suggests that in US10-expressing cells HLA-B*15:03 peptide loading takes place without the quality control exerted by the PLC.

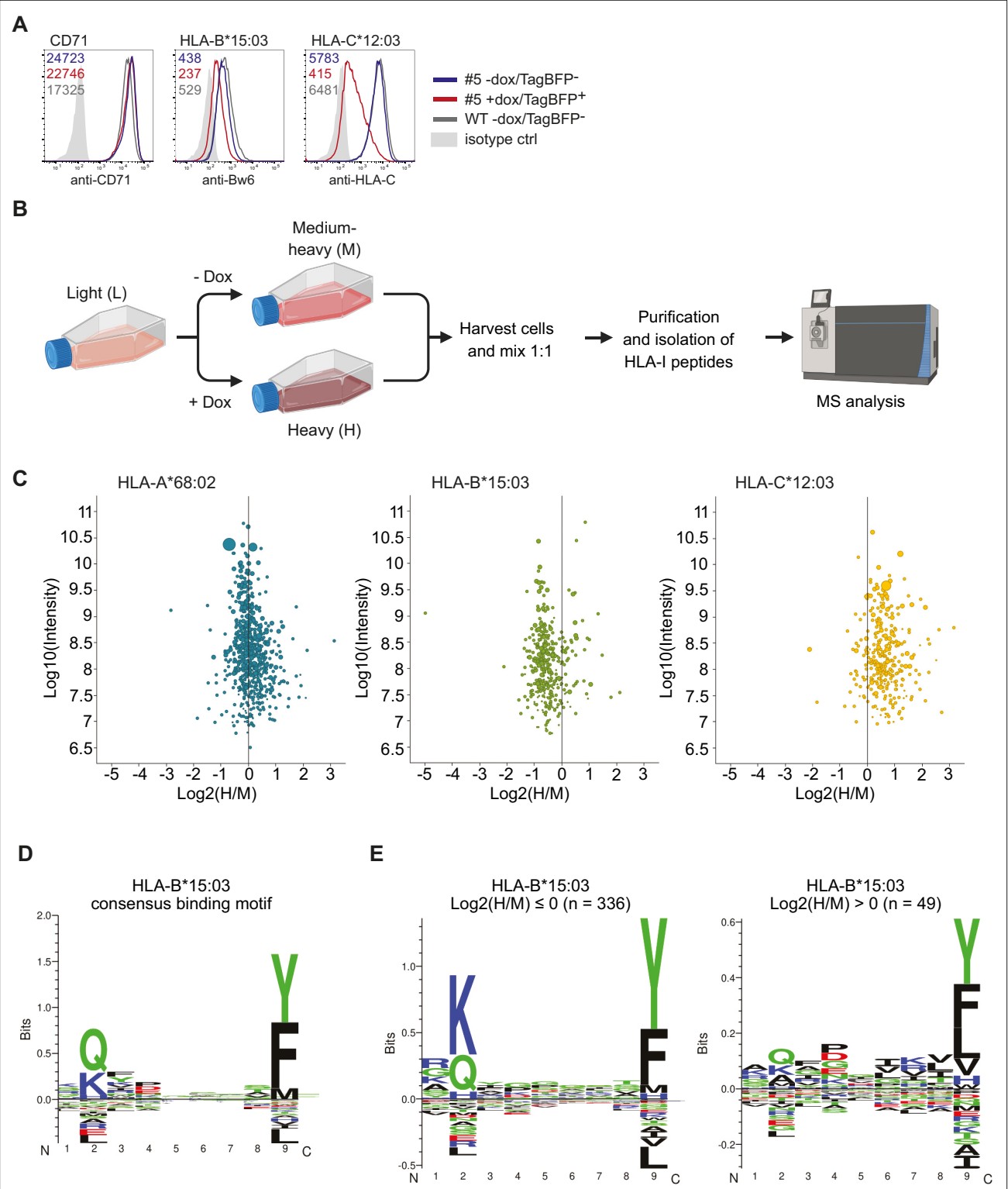

**Figure 5.** Quantitative human leucocyte antigen class I (HLA-I) ligandome analysis confirms genotype-dependent effects by US10. (**A**) Wild-type HeLa cells or TagBFP-T2A-US10i clone #5 were treated with doxycycline (0.33 μg/mL) or DMSO for 24 hr. Subsequently. cells were stained by anti-CD71, anti-Bw6, or anti-HLA-C antibodies and a flow cytometry analysis was performed (cells were gated according to TagBFP expression, TagBFP+ or TagBFP-). (**B**) Experimental scheme of pSILAC immunopeptidomics. HeLa TagBFP-T2A-US10i #5 was pulse-labeled with medium-heavy (M) or heavy (H) amino acids in the presence of DMSO or doxycycline, respectively. After 24 hr, metabolically labeled cells were harvested and combined in a 1:1 ratio. HLA-I peptides were isolated and analyzed by nanoLC-MS/MS. The scheme was created with BioRender.com (**C**) Scatter plots showing median log2 H/M

*Figure 5 continued on next page*

*Figure 5 continued*

ratios and log10 intensity of quantified HLA-I peptides with respect to their HLA-I allele. Dot size correlates with number of ratios used to calculate the corresponding peptide ratio. (**D**) HLA-I consensus binding motif of HLA-B*15:03 obtained from NetMHCpan 4.1 motif viewer (https://services.healthtech.dtu.dk/services/NetMHCpan-4.1/) (**E**) Gibbs clustering analysis (GibbsCluster 2.0) for quantified HLA-B*15:03 peptides with median log2 H/M ratios ≤0 or >0.

The online version of this article includes the following source data and figure supplement(s) for figure 5:

**Figure supplement 1.** Quantitative human leucocyte antigen class I (HLA-I) ligandome analysis confirms genotype-dependent effects by US10.

**Figure supplement 1—source data 1.** Immunoprecipitations in *Figure 5—figure supplement 1B*.

**Figure supplement 1—source data 2.** Immunoprecipitations in *Figure 5—figure supplement 1C*.

## Downregulation of overlapping US10 and US11 transcripts in HCMV-infected cells rescues HLA-I interaction with the PLC

In cells infected with HCMV, several inhibitors of HLA-I are expressed that affect HLA-I expression and function in multiple ways. Studying US10 in this context may reveal its role among these multiple HLA-I inhibitors and their collective impact on antigen presentation. Therefore, we intended to treat MRC-5 fibroblasts with US10 targeting siRNA. However, previous northern blot analysis reported two mRNAs transcribed from the *US11/US10* genome unit (*Jones and Muzithras, 1991*): a long mRNA starting upstream of the *US11* ORF encoding both US10 and US11, and a short mRNA starting in between *US11* and *US10*, which comprises only the *US10* ORF. Hence, application of US10 siRNA could affect US11 expression.

To reassess US10 and US11 transcription, we extracted *US10* and *US11* data from a recent meta-analysis that applied high-throughput sequencing techniques (*Jürges et al., 2022*). In this analysis, two transcription start site (TSS) profiling approaches, dRNA-seq (*Sharma and Vogel, 2014*; *Whisnant et al., 2020*) and STRIPE-seq (*Policastro et al., 2020*), were combined with metabolic RNA labeling (*Erhard et al., 2019*; *Jürges et al., 2018*). Within the *US11/US10* genome unit (*Figure 6—figure supplement 1A*), we found a single polyadenylation (poly-A) signal directly downstream of the *US10* ORF and two active TSSs located directly upstream of the *US10* and the *US11* ORFs, both having an upstream canonical TATA box at the expected distance (*US10* TSS: 31 bp, *US11* TSS: 28 bp; *Figure 6—figure supplement 1A*). In combination with the single poly-A signal, these TSSs are consistent with the short and long mRNAs found by northern blots. However, the profiles of US10 and US11 TSSs, their transcription activity, and protein content in HCMV-infected cells (*Figure 6—figure supplement 1B*) suggest that US10 and US11 are translated from distinct mRNAs.

To test if US10 siRNA could affect US11 expression, we performed a qRT-PCR analysis of RNA isolated at 48 hr p.i. from HCMV-infected cells treated with control or US10 siRNA (*Figure 6A*; US10 siRNA efficacy is demonstrated in *Figure 6—figure supplement 2*). Indeed, the level of *US10* mRNA in US10 siRNA-treated cells was reduced by 92% and *US11* by 67%, verifying that US11 is expressed from a transcript including the *US10* sequence.

Since antibodies that detect US10 and US11 in HCMV-infected cells are not available, for verification of the effect of US10 siRNA on target proteins, we took advantage of BJ-5ta fibroblasts that overexpress HLA-G. Both US10 and US11 can be co-immunoprecipitated with the mAb W6/32 from HCMV-infected BJ-5ta-HLA-G cells. Therefore, these cells were treated with control or US10 siRNA and infected with BAC2 (AD169VarL-derived BAC mutant that lacks the genes *US2-US6*; *Le Trilling et al., 2020*; *Le et al., 2011*) and co-immunoprecipitation with W6/32 was performed. Confirming the RNA analysis, both US10 and US11 proteins were reduced in the presence of US10 siRNA (*Figure 6B*). Hence, in this experimental setup, it is essential to consider the potential effects on HLA-I exerted by both proteins.

Using the same siRNA treatments and HCMV infection described as above, MRC-5 cells were metabolically labeled and HLA-I expression (W6/32 immunoprecipitation) and maturation (EndoH treatment) were analyzed. In accordance with previous observations (*Halenius et al., 2011*), HLA-I molecules were strongly retained both in BAC2 and HB5 (AD169VarS-derived HCMV Δ*US2-US6* BAC mutant) (*Borst et al., 1999*)-infected cells (*Figure 6C*). Treatment with US10 siRNA resulted in an increase of HLA-I, most of which remained EndoH-sensitive. The increased HLA-I expression was

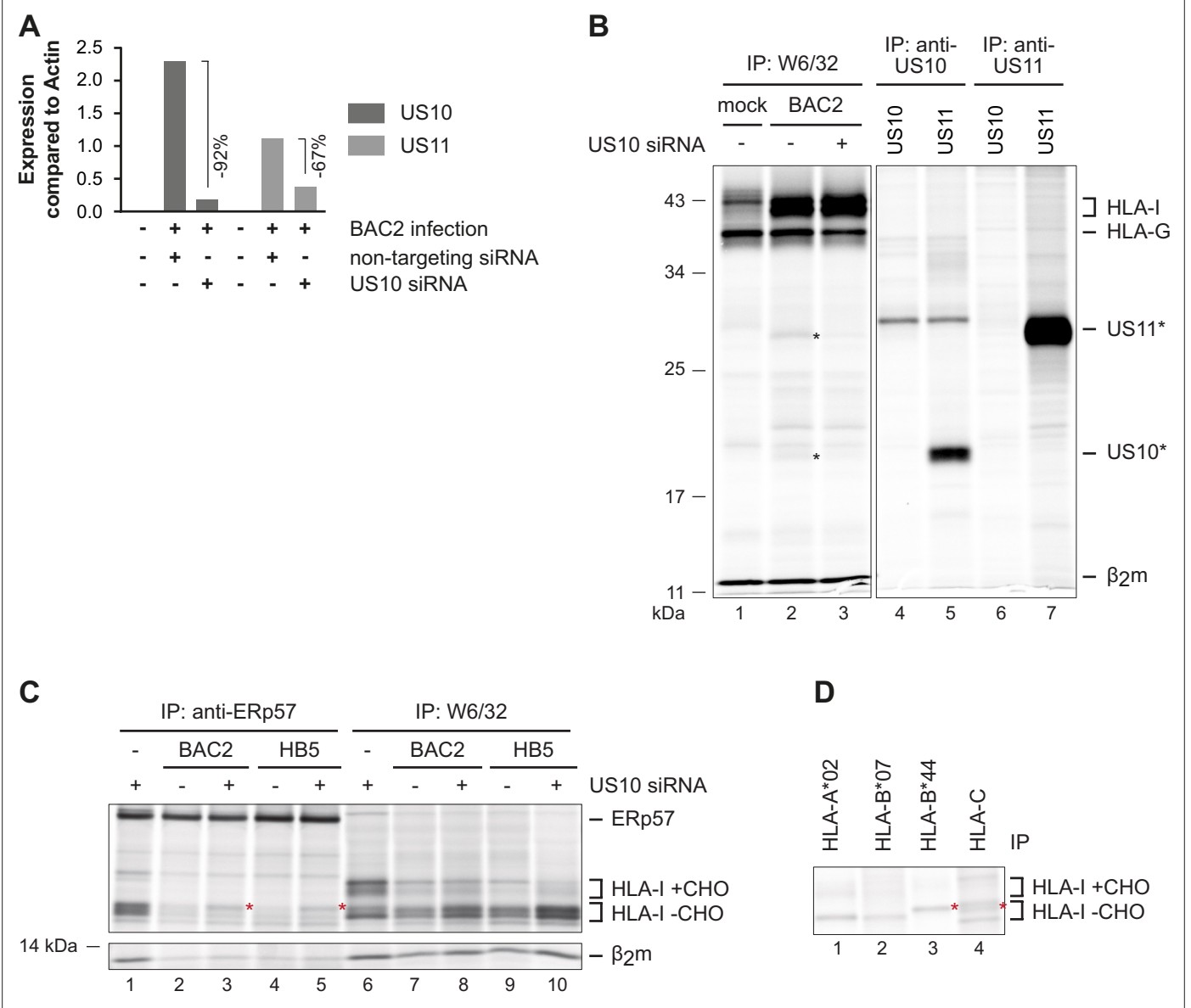

**Figure 6.** Downregulation of overlapping US10 and US11 transcripts in human cytomegalovirus (HCMV)-infected cells rescues human leucocyte antigen class I (HLA-I) interaction with the peptide loading complex (PLC). (**A**) MRC-5 fibroblasts were nucleofected with US10-specific or non-targeting siRNA 24 hr prior to mock treatment or infection with HCMV ΔUS2-6 mutant BAC2 at an MOI (multiplicity of infection) of 5. At 24 hr p.i., RNA was isolated. Subsequently, cDNA was generated and analyzed by qPCR. The binding sites for the used primers are depicted in *Figure 6—figure supplement 2*. Expression of US10 and US11 is shown compared to expression of actin. (**B**) HLA-G-expressing BJ-5ta fibroblasts were treated with siRNA and infected as in (**A**). At 48 hr post-infection, cells were metabolically labeled for 2 hr. Digitonin cell lysates were prepared, and immunoprecipitations were performed as indicated (lanes 1–3). In parallel, immunoprecipitations were performed with HeLa cells transfected with US10 or US11 expression plasmids, lanes 4–7 (*Figure 6—source data 1*). (**C**) MRC-5 fibroblasts were nucleofected with US10-specific or non-targeting siRNA 24 hr prior to mock treatment or infection with the HCMV ΔUS2-6 mutants BAC2 or HB5 at an MOI of 7. Proteins were metabolically labeled at 24 h p.i. for 2 hr, and immunoprecipitations using anti-ERp57 or W6/32 were performed. All samples were treated by EndoH (*Figure 6—source data 2*). Asterisk: strongly increased HLA-I HC when applying US10 siRNA. One of two independent experiments (biological replicates) is shown. (**D**) BAC2-infected MRC-5 fibroblasts were treated as in (**C**), and HLA-I-specific immunoprecipitations were performed as indicated. All samples were EndoH-treated prior to separation by SDS-PAGE (*Figure 6—source data 2*).

The online version of this article includes the following source data and figure supplement(s) for figure 6:

**Source data 1.** Immunoprecipitation in *Figure 6B*.

**Source data 2.** Immunoprecipitations in *Figure 6C and D*.

**Figure supplement 1.** Overlapping US10 and US11 transcripts in human cytomegalovirus (HCMV)-infected cells.

*Figure 6 continued on next page*

*Figure 6 continued*

**Figure supplement 1—source data 1.** Western blots in *Figure 6—figure supplement 2*.

**Figure supplement 1—source data 2.** Immunoprecipitation in *Figure 6—figure supplement 2*.

**Figure supplement 2.** Downregulation of overlapping US10 and US11 transcripts in human cytomegalovirus (HCMV)-infected cells rescues human leucocyte antigen class I (HLA-I) interaction with the peptide loading complex (PLC).

**Figure supplement 2—source data 1.** Uncropped blots from *Figure 6—figure supplement 2B*.

**Figure supplement 2—source data 2.** Uncropped version of *Figure 6—figure supplement 2E*.

probably the result of reduced US11 expression. In addition, a selective effect on the HLA-I molecules in the PLC (anti-ERp57 co-immunoprecipitation) after US10 siRNA treatment was observed; only the intensity of the HLA-I HCs with a slower migration was increased. Since HLA-B*44:02 and HLA-C migrate slower than HLA-A*02:01 and -B*07:02 (*Figure 6D*), this finding suggests that the US10 siRNA has a stronger effect on the recruitment of HLA-B*44:02 and HLA-C molecules to the PLC. This fits well with the observation that in particular HLA-B*15:03/-C*12:03 were excluded from the PLC in US10-expressing HeLa cells (*Figure 2C*) and suggests that also in HCMV-infected cells US10 blocks interaction of specific HLA-I with the PLC and has the potential to skew the HLA-I ligandome.

## US10 siRNA treatment of HCMV-infected cells induces HLA-B antigen presentation

Next, HLA-I cell surface expression was analyzed by flow cytometry on HCMV-infected fibroblasts that were pretreated with US10 or control siRNA. Curiously, no strong effects on HLA-A allotypes were observed, suggesting that the low level of US11 expression is sufficient to control HLA-A (*Figure 7*, *Figure 7—figure supplement 1*). On the contrary, HLA-B allotypes were significantly induced on the cell surface of BAC2-infected fibroblasts following treatment with US10 siRNA. This trend was also observed for HLA-C and -E. In wild-type AD169VarL-infected cells all effects by the siRNA treatment were lost.

To control whether the change in HLA-B expression is important for CD8[+] T-cell recognition, we measured the level of CD8[+] T-cell activation after co-culturing HCMV-specific CD8[+] T-cells with HCMV-infected fibroblasts. As expected, the US10 siRNA treatment did not affect CD8[+] T-cell recognition of AD169VarL-infected cells (*Figure 7C and D*). In contrast, both HLA-B*07:02[pp65]-specific ex vivo expanded CD8[+] T-cells (*Figure 7C*, *Figure 7—figure supplement 2A*) and an HLA-B*08:01[IE1]-specific CD8[+] T-cell clone (*Figure 7D*, *Figure 7—figure supplement 2B*) showed a twofold increase in IFNγ and TNFα expression when co-cultured with BAC2-infected fibroblasts that were pretreated with US10 siRNA. These data emphasize the importance of the *US11/US10* transcript unit in controlling antigen presentation during HCMV infection and prompt future studies to dissect distinct and combined effects of US10 and US11 during HCMV infection.

## Discussion

In HCMV infection, HLA-I antigen presentation is targeted by multiple immunoevasins, employing diverse mechanisms in a coordinated temporal fashion. They control HLA-I geno- and allotype-specific functions across various cell types and environmental conditions (*Ahn et al., 1997*; *Gabor et al., 2020*; *Halenius et al., 2015*; *Hengel et al., 1995*; *Hengel et al., 1997*; *Jones and Sun, 1997*; *Jones et al., 1996*; *Lehner et al., 1997*; *Park et al., 2004*; *Prod'homme et al., 2012*; *Wiertz et al., 1996*; *Zimmermann et al., 2019*). We have recently shown an explicit targeting of HLA-A locus products by US11, suggesting a co-evolutionary relationship (*Zimmermann et al., 2024*), while HLA-B escapes US11-mediated degradation (*Zimmermann et al., 2019*). Here we show that also US10 targets HLA-I molecules in a profound genotype-dependent manner, underscoring the well-nuanced control of HLA-I antigen presentation by HCMV.

## HLA-I downregulation by US10 is HLA-I geno- and allotype-specific

To investigate the specificity of US10 across different HLA-I molecules, we selected a large panel of HLA-I proteins including classical and non-classical HLA-I. Our findings revealed distinct preferences of US10 for specific HLA-I genotypes: tapasin-dependent HLA-B exhibited strong sensitivity, but also

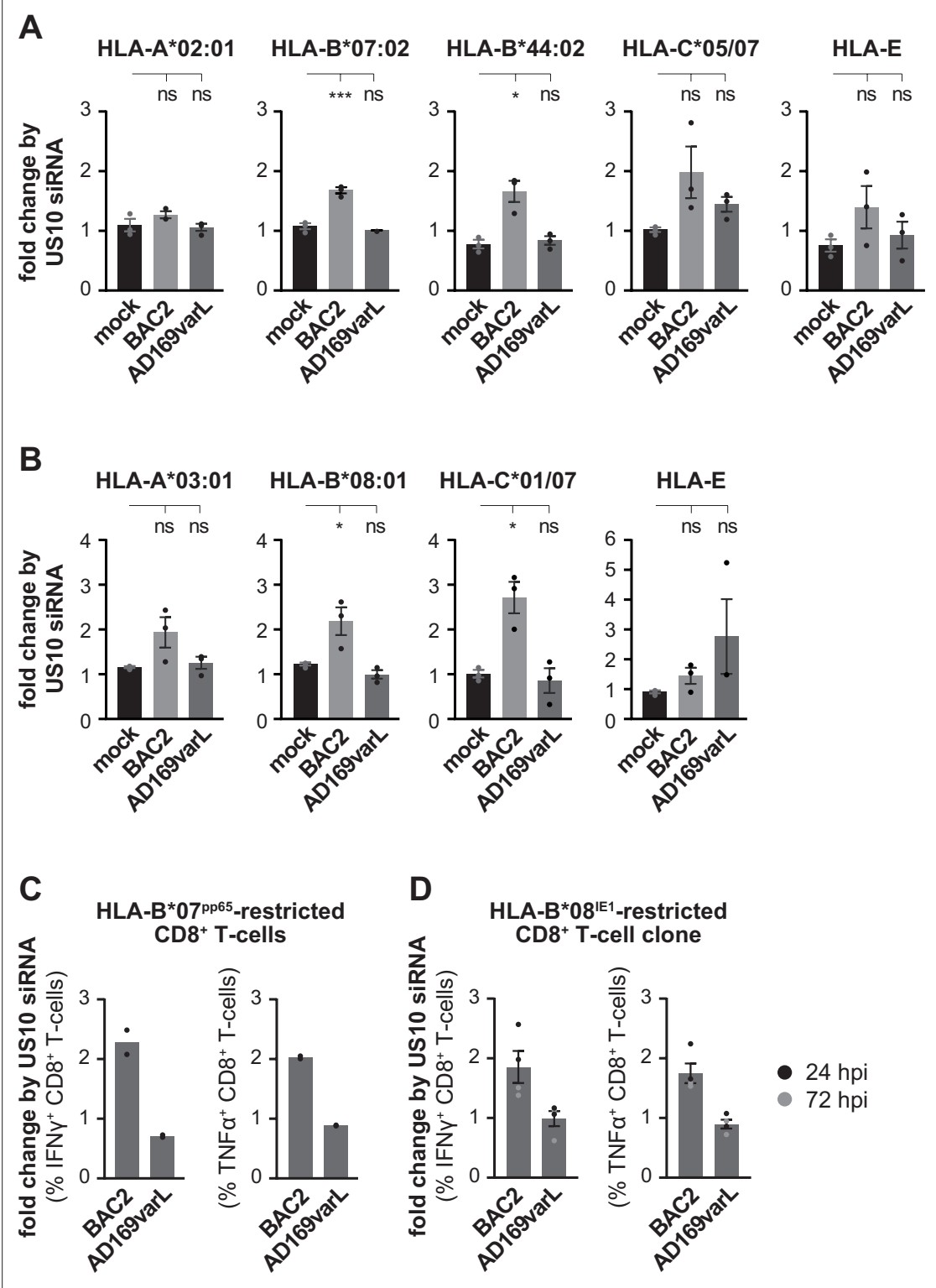

**Figure 7.** US10 siRNA treatment of human cytomegalovirus (HCMV)-infected cells has little effect on HLA-A, but induce HLA-B1027 antigen presentation. (**A, B**) MRC-5 (A) or HF99/7 (B) fibroblasts were nucleofected with US10-specific or non-targeting siRNA 24 hr prior to mock treatment or infection at an MOI of 5 with HCMV ΔUS2-6 mutant BAC2 or with AD169varL. At 48 h p.i., HLA-I surface expression was measured by flow cytometry using antibodies as indicated. Fold change by US10 was calculated as the ratio of the median fluorescence intensity (MFI) of cells treated with US10 siRNA compared to NT-treated cells. Dots represent individual values and bars mean values ± SEM from three independent experiments (biological replicates). Significance was calculated using one-way paired ANOVA followed by Dunnett's multiple comparison test. (**C, D**) HFFα (**C**) or HFF99/7 (**D**)

*Figure 7 continued on next page*

*Figure 7 continued*

fibroblasts were treated and infected as in (**A, B**). At 24 h p.i., the fibroblasts were co-cultured for 5 hr with HLA-B*07:02pp65-specific polyclonal CD8+ T-cells gained from PBMCs (peripheral blood mononuclear cells) (C) or with an HLA-B*08IE1-specific CD8+ T-cell clone at an E/T ratio of 3:1 (**C**) and 5:1 (**D**), respectively. Activation of CD8+ T-cells was determined by intracellular IFNγ and TNFα stain. The percentage of IFNγ- or TNFα-expressing CD8+ T-cells was measured and the fold change by US10 siRNA was calculated. Dots represent individual values from two (**C**) and four (**D**) independent experiments (biological replicates). Co-culturing for (**D**) took place 24 (black dots) or 72 hr p.i. (gray dots). Representative dot plots are shown in *Figure 7—figure supplement 2A and B*.

The online version of this article includes the following figure supplement(s) for figure 7:

**Figure supplement 1.** Human leucocyte antigen class I (HLA-I) analysis of US10 siRNA-treated human cytomegalovirus (HCMV)-infected fibroblasts.

**Figure supplement 2.** CD8+ T-cell activation after co-culture with US10 siRNA-treated human cytomegalovirus (HCMV)-infected fibroblasts.

HLA-G, -E, and most -C showed reduced cell surface expression. In contrast, US10 had no discernible impact on HLA-A allotypes.

The absence of cell-surface regulation of HLA-A was not due to lack of US10 binding. The conformation-independent binding of the anti-HA antibody to HA-tagged HLA-I demonstrated a remarkably conserved binding capacity of US10 to all tested HLA-I HCs (HLA-A, -B, -C, -E, and -G). However, consistent with unaffected cell surface expression of HLA-A allotypes, maturation of endogenous HLA-A*68:02 and transiently expressed HLA-A*02:01 was only modestly delayed in HeLa cells expressing US10.

## Functional analysis of the PLC reveals HLA-I genotypic differences in US10 targeting

A strong correlation between US10-mediated inhibition of HLA-B allotypes and their dependency on tapasin was noted. In addition, HLA-I recruitment to the PLC was hindered by US10. HLA-I ligandome analysis verified that US10 impaired the quality control of HLA-B*15:03 and narrowed its ligandome. This underlines the observation that US10 blocks HLA-B interaction with the PLC and prohibits the quality control exerted by tapasin. In contrast to the effects on HLA-B*15:03, block of PLC recruitment was less pronounced for HLA-A*68:02. Also, no marked effect on the HLA-A*68:02 ligandome was measured after induction of US10 expression.

Interestingly, co-immunoprecipitation experiments indicated US10 interaction with the PLC, which was dependent on the expression level of tapasin and to a lesser extent on HLA-I. This suggests that US10 could bind tapasin directly and block the interaction between HLA-I and tapasin, potentially contributing to the observed inhibition of HLA-I recruitment to the PLC. Another reason for the impaired interaction between HLA-I and the PLC could involve the sequestration of HLA-I by US10 prior to its entry into the PLC. This is supported by the finding that US10 was able to induce further downregulation of sensitive HLA-I in tapasin-deficient cells (*Figure 3C*). Hence, these models are not mutually exclusive. On the contrary, our data suggest that US10 has multitasking abilities and employs several strategies to target HLA-I.

## US10 is able to retain assembled HLA-C and HLA-G in the ER, but not HLA-A and HLA-B

The unexpected finding that US10 associates selectively with HLA-C and -G in their β2m-dimerized form suggests that US10 has evolved an independent molecular targeting strategy for HLA-C and -G. This prominent interaction could explain why downregulation of HLA-C allotypes did not correlate with their level of tapasin dependency. Still, whether tapasin is involved as a co-factor in retention and accumulation of peptide-loaded HLA-C in the ER needs clarification.

Interestingly, forced ER localization of assembled HLA-A*02:01 and HLA-B*07:02 by brefeldin A increased US10 binding to these molecules, demonstrating that this interaction is possible but short-lived. Hence, under normal conditions US10 interaction with assembled HLA-A*02:01 and -B*07:02 does not impede their transport out of the ER. This indeed shows that US10 interaction with HLA-G and -C is qualitatively different from the interaction with HLA-A and -B.

It is surprising that, despite our extensive studies, we did not observe destabilization of HLA-G or any other HLA-I molecule, even though previous research reported that US10 facilitates the degradation of HLA-G (*Park et al., 2010*). Supporting our initial observations in transfected HeLa cells,

treatment with US10 siRNA in HCMV-infected BJ-5ta-HLA-G cells decreased HLA-G levels compared to other HLA-I HCs in the sample (*Figure 6—figure supplement 2*). We observed that modifications to the US10 protein significantly affect its stability and intracellular distribution, potentially causing abnormal effects on HLA-I when the sequence is altered. Therefore, throughout this study the use of modified (epitope-tagged) versions of US10 was avoided. To clarify still elusive features of the US10 protein, in particular with regard to the control of correct ER insertion and folding, further studies are required.

## Downregulation of overlapping US10 and US11 transcripts in HCMV-infected cells has little effect on HLA-A, but induces HLA-B antigen presentation

Previous studies demonstrated an impaired interaction between HLA-I and the PLC in HCMV-infected cells. However, a specific HCMV-encoded factor could not be assigned to this phenotype (*Halenius et al., 2011*). Here we show that US10 most likely contributes to the inefficient HLA-I recruitment to the PLC in HCMV-infected cells. However, with the use of US10 targeting siRNA in these experiments a clear segregation between US10- and US11-mediated effects in HCMV-infected cells was not possible due to the reduction of both proteins. We confirmed previous findings (*Jones and Muzithras, 1991*), demonstrating that US11 is encoded by an mRNA that comprises the *US10* sequence, whereas a separate mRNA species encodes US10 independently, lacking the *US11* sequence. In support of this, two active TSSs directly upstream of the *US10* and the *US11* ORFs were identified and both ORFs were found to share a single polyadenylation signal directly downstream of the *US10* ORF.

Considering the strong effects previously observed with the BAC2-derived *US11*-deletion mutant (*Lübke et al., 2020*; *Zimmermann et al., 2019*), it was surprising that the US10 siRNA treatment of HCMV-infected cells had little effect on HLA-A expression. Possibly, even at low concentrations US11 may be able to control HLA-A expression. In contrast, all studied HLA-B allotypes were increased upon US10 siRNA treatment of BAC2-infected cells, and this resulted in improved antigen presentation as measured by increased CD8+ T-cell responses against HLA-B*07:02[pp65] and HLA-B*08:01[IE1] peptide–MHC-I complexes. Exploring the distinct contribution of each ORF, as well as possible synergistic or antagonistic effects of US10 and US11 on antigen presentation, warrants further attention.

## A model for US10-mediated geno- and allotype-specific HLA-I regulation

US10 is able to bind to all HLA-I molecules early after their synthesis and prior to dimerization with $\beta_2$m (*Figure 8*). Future studies have yet to reveal whether this represents a transient interaction followed by a selective re-encountering at later stages during the conformational maturation of HLA-I

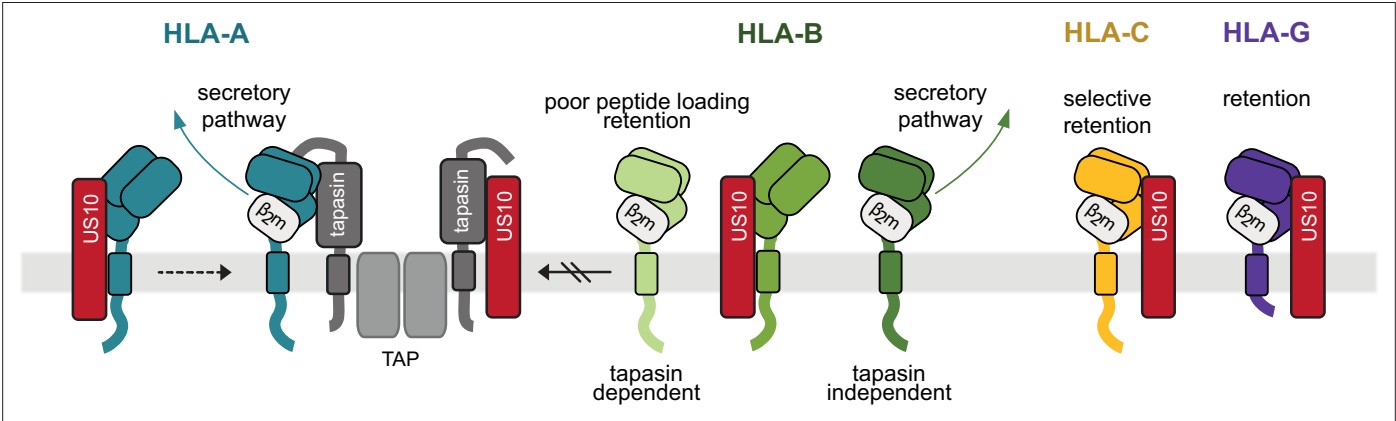

**Figure 8.** Model of human leucocyte antigen class I (HLA-I) geno- and allotype-dependent targeting by US10. US10 (red) is able to bind to all HLA-I heavy chains (HCs) early after their synthesis and prior to dimerization with $\beta_2$m. HLA-A (blue) and HLA-B (green colors) molecules can escape from US10 by dimerization with $\beta_2$m. In addition, US10 blocks HLA-I recruitment to the peptide loading complex (PLC). This has a pronounced inhibitory effect on tapasin-dependent HLA-B allotypes. HLA-A allotypes can overcome this inhibition. $\beta_2$m-assembled HLA-G (purple) and some -C (yellow) molecules are strongly retained in the endoplasmic reticulum (ER).

or whether US10 'co-matures' with selected HLA-I molecules. HLA-A and HLA-B molecules escape early from US10; $\beta_2$m dimerization with the HC promotes loss of binding to US10 and transport out of the ER. This change in affinity could be due to the US10-HC interaction site being partially blocked by $\beta_2$m. In addition, US10 prohibits HLA-I recruitment to the PLC, possibly by interacting directly with tapasin. This leads to a pronounced inhibitory effect on tapasin-dependent HLA-B allotypes, while HLA-A allotypes overcome this inhibition. The remaining interaction between HLA-A and the PLC may be sufficient to support proper peptide loading of HLA-A in our experimental setup.

HLA-G and some -C molecules are strongly retained in a $\beta_2$m-assembled and peptide-loaded state in the ER. The observed US10-mediated retention seems to exert a stronger influence than the absence of control by the PLC. Nevertheless, it is conceivable that the PLC contributes to the retention. Future studies will provide more insights into the mechanistic and the immunological role of US10 in modulating HLA-C and -G. Given the relevance of congenital HCMV infections and the fact that trophoblasts do not express HLA-A and -B, but HLA-C, -G, and -E (*Hackmon et al., 2017*), this prompts the question as to whether US10 makes an important pathogenic contribution to placental infection, subsequent transmission to the fetus. and ultimate disease of the congenitally infected newborn.

## Materials and methods
### Molecular cloning
MHC-I and MHC-I like molecules were cloned into a Tpn-SP-pIRES-EGFP vector via PstI or NsiI and BamHI. This vector encodes the tapasin signal peptide and an HA-tag N-terminally to the insertion site for the MHC-I sequence. The HA-tagged constructs where then subcloned into the puc2CL6IP vector using NheI and BamHI (all primers in *Table 1*), which was used for expression. Sequences for HLA-I molecules were obtained from different sources: HLA-C*04:01 and -G*01:01 from cDNA prepared from JEG-3 cells (ATCC HTB-36); HLA-A*24:02, -C*05:01, and H-2K$^b$ were ordered as gBlock gene fragments (Integrated DNA Technologies); HA-MR1 was purchased from the pcDNA3.1 vector (BioCat); HA-tagged MICA*004 was obtained from a previously published construct (*Seidel et al., 2015*). The sequence for HLA-E*01:01 was obtained as a gBlock gene fragment. To add the signal peptide of HLA-A*02:01, it was included into a primer. RL8, US10, US10HA, UL40, and US9HA were amplified from AD169 HCMV cDNA and cloned into pIRES-EGFP or puc2CL6IP via NheI/BamHI and XhoI/BamHI (for US2HA), respectively. TagBFP-T2A-US10, a fusion construct of the blue fluorescent protein TagBFP in frame with the self-cleaving T2A peptide and US10, was generated by gene synthesis (BioCat) and inserted into pSBtet-Pur (a kind gift from Eric Kowarz; Addgene plasmid #60507) (*Kowarz et al., 2015*) via SfiI. Constructs not mentioned here have been previously described (*Beutler et al., 2013*; *Zimmermann et al., 2019*). For mutation of HLA-I gRNA target sites, the Q5 site-directed mutagenesis kit (New England BioLabs) was applied. A wild-type tapasin sequence was cloned into NheI/EcoRI sites of pcDNA3.1.

### Cell culture, transfection, nucleofection, and generation of stable cell lines
HeLa cells (human, ATCC CCL-2), MRC-5 fibroblasts (human, ATCC CCL-171), BJ-5ta (hTERT; human, ATCC-CRL-4001), and HFFα human foreskin fibroblasts (a kind gift from Dieter Neumann-Haefelin and Valeria Kapper-Falcone, Institute of Virology, Medical Center University of Freiburg, Freiburg, Germany) were grown in DMEM (Life Technologies) supplemented with 10% (v/v) FCS (Biochrom, Sigma-Aldrich or PAN-Biotech) and 1% (v/v) penicillin/streptomycin (Life Technologies, stock: 5000 U/mL) at 37°C and 5% $CO_2$. Mycoplasma-negative cell cultures were verified by PCR every 2–3 wk. For transient expression, HeLa cells were transfected with SuperFect (QIAGEN) for 20 hr and fibroblasts cells were nucleofected using the SE Cell Line AD-Nucleofector Kit (Lonza). Knockdown procedures were performed with siRNA purchased from Riboxx (non-targeting siRNA: UUGUACUACACAAAAG UACCCCC; US10 targeting #1: UUCUGAAUAACACAGCCGCCCCC) using the SE Cell Line Kit (Lonza) for fibroblast transfection and Lipofectamine RNAiMax (Invitrogen) for HeLa cells.

HeLa cells stably expressing US9HA (control cells), US10, or US10HA and BJ-5ta stably expressing HA-HLA-G*01:01 were generated by lentiviral transduction. A tapasin knockout cell line was generated by transient expression of SpCas9 and gRNA targeting the sequence GCCCTATACGCCAGGC

**Table 1.** Primer sequences for molecular cloning.

| | |
|---|---|
| HA-HLA-A*24:02 | 1. CGTATGCATTAGGAGGCTCCCACTCCATGAGG<br>2. CGAGGATCCTCACACTTTACAAGCTGTGAGAGAC |
| HA-HLA-C*04:01 | 1. CGAATGCATTAGGAGGCTCCCACTCCATGAGG<br>2. CGTGGATCCTCAGGCTTTACAAGCGATGAGAG |
| HA-HLA-C*05:01 | 1. CGTATGCATTAGGATGCTCCCACTCCATGAGG<br>2. CGAGGATCCTCAGGCTTTACAAGCGATGAGAG |
| HA-HLA-G*01:01 | 1. CGAATGCATTAGGAGGCTCCCACTCCATGAGG<br>2. CGTGGATCCTCAATCTGAGCTCTTCTTCCTCCAC |
| HA-H-2K[b] | 1. CGTATGCATGGCCCACACTCGCTGAGG<br>2. CGAGGATCCTCACGCTAGAGAATGAGGG |
| HA-MICA*004 | 1. CGTGCTAGCGCCGCCACCATGGG<br>2. CGTGGATCCCTAGGCGCCCTCAGTGG |
| HLA-[A2]E*01:01 | 1. CGTGCTAGCATGGCCGTCATGGCGCCCCGAACCCTCGTCCTGCTACTCTCGGGGGCTCTG<br>GCCCTGACCCAGACCTGGGCGGGCTCCCACTCCTTGAAGTATTTCC<br>2. CGTGGATCCTTACAAGCTGTGAGACTCAGACC |
| ΔCRISPR2-HA-HLA-A*02:01 | 1. AAGACTATATTGCCCTGAAAGAGGACCTG<br>2. TACCATCGTACGCGTACTGGTGGTACCCG |
| ΔCRISPR2-HA-HLA-B*07:02 | 1. *AAGACTATATTGCCCTGAACGAGGACCTG*[*]<br>2. TACCATCGTACGCGTACTGGTCATGCCCG |
| ΔCRISPR2-HA-HLA-B*44:02 | 1. *AAGACTATATTGCCCTGAACGAGGACCTG*[*]<br>2. TACCATCGTACGCGTCCTGGTCATACCCG |
| ΔCRISPR2-HA-HLA-C*05:01 | 1. AAGACTATATTGCCCTGAATGAGGACCTG<br>2. TACCATCGTACGCGAACTGGTTATACCCG |
| ΔCRISPR2-HA-HLA-C*07:02 | 1. *AAGACTATATTGCCCTGAACGAGGACCTG*[*]<br>2. TACCATCGTACGCGGACTGGTCATACCCG |
| ΔCRISPR2-HA-HLA-E*01:01 | 1. AAGACTATCTTACCCTGAATGAGGACCTG<br>2. TACCATCGTACGCGAACTGTTCATACCCG |
| ΔCRISPR2-HA-HLA-G*01:01 | 1. AAGACTATCTTGCCCTGAACGAGGACCTG<br>2. TACCATCGTACGCATACTGTTCATACCCGC |
| RL8 | 1. CGGGCTAGCATGCCTCACGGCCATCTC<br>2. GCAGGATCCTCAGCTAAAAACAGCGGACAGTC |
| US10 | 1. CGGGCTAGCATGCTACGCCGGGGAAGC<br>2. GCCGGATCCTTATTCGCGAGGTGGATAATAACCG |
| US10HA | 1. CGGGCTAGCATGCTACGCCGGGGAAGC<br>2. GCCGGATCCTCATGCGTAATCTGGAACATCGTATGGGTATTCGCGAGGTGGATAATAA CCG |
| US2 | 1. CGTCTCGAGATGAACAATCTCTGGAAAGCCTG<br>2. GCAGGATCCTCAGCACACGAAAAACCGCAT |
| US2HA | 1. CGTCTCGAGATGAACAATCTCTGGAAAGCCTG<br>2. GCAGGATCCTCATGCGTAATCTGGAACATCGTATGGGTATGCACACGAAAAACCGCATCC |
| US3HA | 1. CGAGCTAGCATGAAGCCGGTGTTGGTG<br>2. CGTGGATCCTTACGCGTAATCTGGAACATCGTATGGGTAAATAAATCGCAGACGGGCG |
| US9HA | 1. CGGGCTAGCATGATCCTGTGGTCCCCG<br>2. GCCGGATCCTCATGCGTAATCTGGAACATCGTATGGGTAATCGTCTTTAGCCTCTTCTTCC |
| UL40 | 1. GCAGCTAGCGCCGCCACCATGAACAAATTCAGCAACACTCG<br>2. CGAGGATCCTCAAGCCTTTTTCAAGGCG |

[*]Primers written in italics are identical.

CTGG using plasmids kindly gifted by J. Keith Joung (Addgene plasmids #43861 and #43860) as detailed elsewhere (**Müller et al., 2016**). The HLA-I knockout HeLa cells were generated as previously described (**de Waard et al., 2021**). HeLa HLA-I and $\beta_2$m double knockout cells were generated by nucleofection of SpCas9 and guide RNA ribonucleoprotein complexes (Synthego) into HLA-I HeLa

knockout cells. For generation of a doxycycline-inducible US10-expressing HeLa cell line (TagBFP-T2A-US10i), a sleeping beauty transposon system was used. Cells were transfected with pCMV(CAT) T7-SB100 (a kind gift from Zsuzsanna Izsvak; Addgene plasmid #34879; *Mátés et al., 2009*) and pSBtet-Pur, encoding TagBFP-T2A-US10. Subsequently, the cells were treated by puromycin (Sigma-Aldrich) and single-cell clones were selected.

## Viruses

Generation, reconstitution, and propagation of the HCMV strain AD169VarL and the recombinant HCMV ΔUS2-6 mutants BAC2 (GenBank accession number MN900952.1) based on the AD169varL and HB5 based on the AD169VarS (GenBank accession number X17403) strains, were previously described (*Borst et al., 1999*; *Halenius et al., 2011*; *Hengel et al., 1995*; *Le Trilling et al., 2020*; *Le et al., 2011*). In experimental settings, fibroblasts were infected with an MOI (multiplicity of infection) of 5–7 with centrifugal enhancement (800× *g* for 30 min).

## Antibodies

The following antibodies were used in this study: W6/32 (anti-pan-HLA-I assembled with $\beta_2$m and peptide; *Parham et al., 1979*), anti-HLA-B*07:02 (BB7.1; *Brodsky et al., 1979b*), anti-HLA-A*02 (BB7.2; *Brodsky et al., 1979b*), anti-$\beta_2$m (BBM.1; *Brodsky et al., 1979a*), anti-HLA-B*44 (TT4-A20; *Tahara et al., 1990*), anti-HLA-A*03 (GAP A3; *Berger et al., 1982*), anti-HLA-C (DT-9, BioLegend, 373302, RRID:AB_2650941), anti-HLA-E (3D12, BioLegend, 342602, RRID:AB_1659247), anti-HA produced in mouse (Sigma-Aldrich, H3663, RRID:AB_262051) or rabbit (Sigma-Aldrich, H6908, RRID:AB_260070), anti-ERp57 (abcam, ab13506, RRID:AB_1140700), anti-β-actin (Sigma-Aldrich, A2228, RRID:AB_476697), anti-CD28 (BD Biosciences), APC-coupled anti-HA (Miltenyi Biotec, 130-098-404, RRID:AB_2751024), APC-coupled IgG1-isotype control (Miltenyi Biotec, 130-113-200, RRID:AB_2733881), APC-coupled anti-CD8 (BD Biosciences), APC-coupled anti-CD56 (Miltenyi Biotec, 130-113-305, RRID:AB_2726084), APC-coupled goat anti-mouse IgG (BD Biosciences, 550826, RRID:AB_398465), FITC-coupled anti-IFNγ (BD Biosciences, 554551, RRID:AB_395473), PE/Cy7-coupled anti-TNFα (BioLegend, 506323, RRID:AB_2204356), HRP-coupled goat anti-mouse IgG (Dianova, 115-035-146, RRID:AB_2307392), HRP-coupled goat anti-rabbit IgG (Sigma-Aldrich, 12-348, RRID:AB_390191), FITC-coupled anti-Bw6 (Miltenyi Biotec, 130-123-264, RRID:AB_2819460), and APC-coupled anti-CD71 (Miltenyi Biotec, 130-123-788, RRID:AB_2857636). Polyclonal anti-tapasin and anti-US10 were raised in rabbits (GenScript) using synthetic peptides (aa 418–428 and aa 54–67, respectively) and anti-US11 was described previously (*Zimmermann et al., 2019*).

## Flow cytometry

HeLa cells were harvested using trypsin and fibroblasts using accutase (Sigma-Aldrich). The cells were washed and stained with antibodies in PBS with 3% (v/v) FCS. Cells were further stained with DAPI (but not in *Figure 5A* and *Figure 5—figure supplement 1*) before flow cytometry analysis (FACS Canto II [BD Biosciences], FlowJo [Tree Star]). IFNγ treatment was performed with 1000 U/mL (BioLegend) for 16 hr. FcR blocking (FcR blocking reagent, Miltenyi Biotec) and fixation in 4% paraformaldehyde were performed for infected cells. For intracellular staining, Cytofix/Cytoperm from BD Biosciences was applied.

## Immunoprecipitation

Immunoprecipitations were performed as previously described (*Halenius et al., 2011*). Briefly, cells were cultured in six-well plates and metabolically labeled (Easytag Express $^{35}$S-Met-Cys protein labeling mix [PerkinElmer]) with 0.2 mCi/mL for various times. In pulse chase experiments, cells were washed and incubated in DMEM containing additional methionine/cysteine (Sigma-Aldrich). Cells were lysed in 1% (w/v) digitonin (Calbiochem) lysis buffer (140 mM NaCl, 20 mM Tris [pH 7.6], 5 mM $MgCl_2$) containing cOmplete protease inhibitor (Roche). Lysates were incubated with antibodies for 1 hr at 4°C in an overhead tumbler before immune complexes were retrieved by protein G (GE Healthcare or abcam) or A Sepharose (GE Healthcare). Subsequently, the beads were washed with increasing NaCl concentrations. Endoglycosidase H digestion was performed according to the manufacturer's instructions (New England BioLabs). Protein complexes were dissociated at 95°C in sample buffer containing 150 mM DTT prior to loading on a gradient SDS-PAGE. Gels were fixed, dried, exposed to x-ray films

or phosphor screens, and analyzed using Typhoon FLA 7000 (GE Healthcare). For better illustration, contrast and light settings were adjusted in the figures. Band intensities were determined using the ImageQuant TL Software (GE Healthcare Life Sciences). The background signal was subtracted from values used for graphical visualization of quantification.

## Pulsed stable isotope labeling by amino acids in cell culture (pSILAC)

Metabolic labeling by pulsed SILAC (pSILAC) was conducted prior to immunopeptidomic analysis. At the beginning of each pSILAC experiment, the growth medium of the HeLa TagBFP-T2A-US10i cells was changed to DMEM, lacking L-arginine, L-lysine, and L-leucine (Thermo Scientific), supplemented with 10% dialyzed FCS (Capricorn Scientific), 0.5% penicillin/streptomycin, and 200 mg/L L-proline (Merck) as well as either medium-heavy amino acids (87.3 mg/L $^{13}C_6$-L-arginine, 147.6 mg/L $D_4$-L-lysine, 105 mg/L $D_3$-L-leucine) or heavy amino acids (87.3 mg/L $^{13}C_6$,$^{15}N_4$-L-arginine, 147.6 mg/L $^{13}C_6$,$^{15}N_2$-L-lysine, 105 mg/L $^{13}C_6$-L-leucine). All SILAC amino acids were purchased from Cambridge Isotope Laboratories (Eurisotop). To induce US10 expression, doxycycline (Merck, 2 mg/mL solution in DMSO) was added to heavy SILAC medium at a final concentration of 1 µg/mL. As a control, equivalent amounts of DMSO (Merck) were added to the medium-heavy SILAC medium. After 24 hr, the metabolically labeled cells were harvested and combined in a 1:1 ratio, yielding a cell pellet of 100 million doxycycline-treated and 100 million untreated cells.

## Purification and isolation of HLA-I peptides for MS

HLA-I peptides were isolated by immunoaffinity purification as previously described with some minor modifications (*Bernhardt et al., 2022*; *Chong et al., 2018*). In brief, W6/32 antibody (kindly provided by Hans-Georg Rammensee, Department of Immunology, University of Tuebingen, Germany) was incubated with protein A Sepharose (Protein A-Sepharose 4B, Thermo Scientific) beads at a ratio of 2.5 mg per 1 mL bead slurry, using 0.375 mg of antibody and 75 µL sepharose beads per 100 million cells. Cells were lysed in PBS containing 0.25% sodium deoxycholate (Merck), 0.2 mM iodoacetamide (IAA) (Merck), 1 mM EDTA (Merck), 1× cOmplete protease inhibitors (Roche), 1 mM phenylmethylsulfonylfluoride (Santa Cruz Biotechnology), 1% n-octyl-β-D glucopyranoside (Santa Cruz Biotechnology) at 4°C for 1 hr at a concentration of 80 million cells per mL. Lysates were clarified by centrifugation at 16,000×*g* for 20 min at 4°C. A 96-well filter plate (Waters, #186002799) was prepared by washing the wells with 2 mL of acetonitrile (ACN) (Merck), followed by 2 mL of 0.1% trifluoroacetic acid (TFA) (Merck) and 2 mL of PBS buffer (Merck). W6/32-loaded sepharose beads were filled into a well and the clarified lysate was loaded in the well by gravity flow on ice. Wells were washed four times with 2 mL of 150 mM NaCl, 20 mM Tris pH 8 (Merck), then washed four times with 2 mL of 400 mM NaCl, 20 mM Tris pH 8, three times with 2 mL of 150 mM NaCl, 20 mM Tris pH 8, and two times with 2 mL of 20 mM Tris-buffer pH 8. Peptide-HLA-I complexes were eluted with 1 mL of 1% TFA. Immunoaffinity eluates were lyophilized (Alpha 1–2 LDplus, Christ Gefriertrocknungsanlagen GmbH) and HLA-I peptides were isolated by restricted access material solid-phase extraction (RAM-SPE). RAM-SPE columns were prepared by packing a pipette tip with a filter (Whatman, #1822-047, GF/C 47 mm) and 10 mg of RAM material (MAYI-ODS(G), Shimadzu, MZ-Analysentechnik, #228-40835-97). The columns were flushed with 600 µL of 10 mM ammonium acetate buffer pH 7 (Honeywell Fluka). Freeze-dried samples were resuspended in 200 µL of 1.5% ACN in 10 mM ammonium acetate buffer pH 7 and loaded on the RAM-SPE column. After washing with 800 µL of 10 mM ammonium acetate buffer pH 7, MHC-I peptides were eluted with 350 µL of 60% ACN in 10 mM ammonium acetate buffer pH 7 and subsequently lyophilized.

## NanoLC-MS/MS analysis

LC-MS/MS analysis was performed as described previously (*Bernhardt et al., 2022*). HLA-I peptides were dissolved in 2% ACN, 0.1% formic acid (Fisher Chemical), and nanoLC-MS/MS analyses were performed on an Orbitrap Fusion (Thermo Scientific) equipped with a PicoView Ion Source (New Objective) and connected to an EASY-nLC 1000 (Thermo Scientific). Samples were loaded on a trapping column (2 cm × 150 µm ID, PepSep) and separated on a capillary column (30 cm × 150 µm ID, PepSep), both packed with 1.9 µm C18 ReproSil. Peptides were separated with a flow rate of 500 nL/min and a solvent system consisting of 0.1% formic acid (solvent A) and 80% ACN, 0.1% formic acid (solvent B) using the following gradient: 3–20% solvent B in 15 min, 20–45% solvent B in 45 min, and

45–98% solvent B in 5 min. Eluting peptides were analyzed using a higher-energy collisional dissociation (HCD) method, generating spectra from doubly charged peptides, as well as a combined HCD and electron transfer dissociation (ETD) method, generating spectra from singly and triply charged peptides. For both methods, a top speed data-dependent method with a fixed cycle time of 1.5 s and internal calibration using EASY-IC was used. Both MS and MS/MS scans were acquired in the Orbitrap analyzer with a resolution of 60,000. Precursor selection range was set to m/z 751–1601 (singly charged peptides), m/z 376–801 (doubly charged peptides), and m/z 251–534 (triply charged peptides), with a predictive automatic gain control target of 250% ($1 \times 10^6$) and a maximum injection time of 100 ms. Singly and doubly charged precursors were selected for HCD only, whereas triply charged precursors were selected for HCD and for ETD. HCD fragmentation with normalized stepped collision energy (30, 35, 40%) was applied. Dynamic repeat count was set to 1 with an exclusion duration of 8 s. For precursor selection, an intensity threshold of $1 \times 10^4$ was used. For MS/MS spectra, AGC target was set to 100% ($5 \times 10^4$) and maximum injection time to automatic.

## MS data analysis

Identification of MHC-I peptides from LC-MS/MS data was performed as previously described in detail (*Bernhardt et al., 2022*; *Erhard et al., 2020*). De novo peptide sequencing was performed with PEAKS Xpro (Bioinformatics Solutions Inc) (*Zhang et al., 2012*), including oxidation (Met), pyroglutamate formation from N-terminal glutamine, carbamidomethylation (Cys), and the isotope labels for Arg6, Lys4, Leu3 and Arg10, Lys8, Leu6 as variable modifications. For every peptide, a total of six variable modifications were allowed. The top 10 de novo sequencing candidates were reported for each fragment ion spectrum and matched against the three-frame translation of the human transcriptome (ENSEMBL 90) and the six-frame translation of the reference human genome (CRCh38) using Peptide-PRISM. Matched peptides were categorized according to the genomic region in which they are encoded (CDS, in-frame with annotated protein; 5′ UTR; off-frame, located inside, but out-of-frame, of protein-encoding ORF; 3′ UTR; ncRNA; Intronic; Intergenic). HLA peptide binding predictions were performed with NetMHCpan 4.0 (*Jurtz et al., 2017*). A custom fasta database containing all HLA-I peptides identified by Peptide-PRISM at an FDR of < 10% was generated and used to determine H/M ratios for all HLA-I peptides by MaxQuant 2.4.2.0 (*Cox and Mann, 2008*). Digestion mode was set to 'no digestion'. Multiplicity for the SILAC-labeling was set to 3 (light, medium, and heavy) with a maximum of six labels per peptide. Arg6, Lys4, Leu3 were selected as medium labels and Arg10, Lys8, Leu6 as heavy labels. FDR filtering was turned off by setting PSM FDR, protein FDR, and site decoy fraction to 1. Minimum scores for modified and unmodified peptides were set to 25. Finally, the Re-quantify option was used for improving quantification of large ratios. Apart from these adapted settings, the MaxQuant default parameters were used. The MaxQuant evidence table was merged with the Peptide-PRISM results table by peptide sequence. Median log2 H/M ratios were calculated for all peptides from the H/M ratios given in the MaxQuant evidence table. Median lg2 H/M ratios calculated only from H/M ratios of type 'ISO-MSMS' with either missing labeling state 'medium' or 'heavy' were excluded. Gibbs clustering of quantified HLA-B*15:03 peptides was performed with GibbsCluster 2.0 (https://services.healthtech.dtu.dk/services/GibbsCluster-2.0/) using default parameters for MHC class I ligands of length 8–13 (*Andreatta et al., 2013*).

## Western blot analysis

Cell lysates were separated by SDS-PAGE and transferred to a nitrocellulose membrane (Amersham Protan, GE Healthcare). Incubation with specific antibodies was followed by peroxidase-conjugated secondary antibodies and detection using SignalFire ECL Reagent (Cell Signaling Technology) and the ChemiDoc XRS System (Bio-Rad).

## Transcriptional analysis of US10 and US11

Functional genomics data for the US10/11 locus were extracted from our recent integrative meta-analysis (*Jürges et al., 2022*). Briefly, primary human foreskin fibroblasts (HFFα) were infected with HCMV strain TB40 at MOI 10 and two TSS profiling methods (dSLAM-seq [*Sharma and Vogel, 2014*; *Whisnant et al., 2020*], Stripe-seq [*Policastro et al., 2020*]) were applied. All TSS profiling data were analyzed using iTiSS (*Jürges et al., 2021c*), published Ribo-seq data (*Stern-Ginossar et al., 2012*) was analyzed using PRICE (*Erhard et al., 2018*), and protein expression data were directly taken

from the supplementary material of a previous study (*Weekes et al., 2014*). The genome browser to visualize all data sets is available on zenodo (https://doi.org/10.5281/zenodo.5801030), and a web-based tool for generating time-course plots is available on the project website (https://erhard-lab.de/web-platforms).

## Quantitative RT-PCR

RNA was isolated using the NucleoSpin RNA kit from Macherey-Nagel accrrording to the manufacturer's instructions. cDNA was generated using the QuantiTect Reverse Transcription Kit from QIAGEN according to the manufacturer's instruction. The qPCR was performed with a SYBR Green PCR Master mix (Applied Biosystems) and the primers US10-SYBR1 ACGACGGGGAAAATCACGAA, US10-SYBR2 CAGAGTAGTTTCGGGGTCGG, US11-SYBR1 TTGTTCGAAGATCGCCGTCT, US11-SYBR2 AAAA TGTCGGTGCAGCCAAC (*Figure 6—figure supplement 2*) and QIAGEN control primer Hs_ACTB_1_ SG QuantiTect. Expression of US10 and US11 was determined compared to control using the $2^{-\Delta Ct}$ method.

## Generation of HLA-B*07/pp65-specific CD8+ T-cell polyclones

PBMCs from an HCMV-seropositive, healthy, HLA-B*07:02-positive donor (female) were gained from EDTA blood by density separation (Lymphocyte Separation Medium [AnproTech]) and CD8+ T-cells were isolated with the human CD8+ T-cell isolation kit (Miltenyi Biotec) and a MidiMACS LS-column according to the manufacturer's instructions. Isolated CD8+ T-cells ($2 \times 1.5 \times 10^6$ cells) or original PBMCs (peripheral blood mononuclear cells) ($2 \times 2 \times 10^6$ cells) were cultured in RPMI 1640 medium (supplemented with 10% [v/v] FCS (fetal calf serum) [PAN Biotech], 1% [v/v] penicillin/streptomycin [Life Technologies] and 1.5% [v/v] HEPES [Life Technologies]). 0.5 µg/mL anti-CD28 antibody (BD Biosciences) and 5 µM of an HLA-B*07:02-specific pp65 peptide (417-426): TPRVTGGGAM (Genaxxon [purity: 98.3%]) were added to the medium. 50% of the medium was replaced by fresh, supplemented RPMI with 20 IU/mL IL-2 (Stemcell) every 3 d. After 14 d, specificity of the cells was tested by staining with tetramers generated from HLA-I easYmer and streptavidin-PE (BD Biosciences) according to the manufacturer's instructions (immunAware). For analysis, $5 \times 10^4$ cells were incubated with the tetramer, further stained with anti-CD8 (BD Biosciences) and viability dye (EBioscience), washed, fixed in 2% PFA and measured on a FACS Canto.

## Generation of HLA-B*08^IE1-specific CD8+ T-cell clones

PBMCs ($2 \times 10^6$) from HCMV seropositive, HLA-B*08:01-positive donors were cultured in RPMI 1640 medium (Life Technologies; supplemented with 10% [v/v] FCS [PAN Biotech], 1% [v/v] penicillin/streptomycin [Life Technologies], and 1.5% [v/v] HEPES [Life Technologies]). 0.5 µg/mL anti-CD28 antibody (BD Biosciences) and 5 µM of an HLA-B*08:01-specific IE1 peptide (199-207: ELRRKMMYM; Genaxxon [purity: 99.3%]) were added to the medium for 2 wk. Fifty percent of the medium was replaced by fresh, supplemented RPMI with 20 IU/mL IL-2 (Stemcell) every 3 d. After 10 d, specificity of the cells was tested by staining with tetramers generated from HLA-I easYmer and streptavidin-PE (BD Biosciences) according to the manufacturer's instructions (immunAware). $5 \times 10^4$ cells were incubated with the tetramer, further stained with anti-CD8 (BD Biosciences) and viability dye (EBioscience), washed, fixed in 2% PFA, and measured on a FACS Canto or Fortessa LSR. To generate clones, cells were sorted on the FACS Melody. They were cultured in RPMI 1640 medium (supplemented with 10% [v/v] human serum and 1% [v/v] penicillin/streptomycin [Life Technologies]) together with PHA-M (Sigma-Aldrich) and $2 \times 10^6$ feeder cells per mL. Feeder cells were gained from fresh PBMCs (EDTA blood, density separation; Lymphocyte Separation Medium [AnproTech]) and irradiated with 30 Gy for 30 min, Cs-source. Three days after sorting, the clones were fed with RPMI with human serum, 100 IU/mL IL-2 and the abovementioned supplements. Clones were cultured for 14 d before further analysis. The project was approved by the ethical committee at the Albert-Ludwigs-University Freiburg (22-1196).

## Analysis of the activation of HLA-B/CMV-specific CD8+ T-cells by HCMV infection

$1 \times 10^5$ HFFα or HFF99/7 fibroblasts nucleofected with US10- or non-targeting siRNA were seeded in DMEM (Dulbecco's Modified Eagle Medium, Life Technologies) supplemented with 10% (v/v) FCS

(PAN Biotech) and 1% (v/v) penicillin/streptomycin (Life Technologies). After 24 hr, cells were infected with HCMV at an MOI of 5. At 24 or 72 hr p.i., the infected cells were co-cultured with HLA-B/CMV-specific CD8+ T-cells for 5 hr (E:T ratio: 3:1 [HFFα] or 5:1 [HFF99/7]) in RPMI 1640 medium (Life Technologies)(supplemented with 10% [v/v] FCS [PAN Biotech], 1% [v/v] penicillin/streptomycin [Life Technologies], and 1.5% [v/v] HEPES [Life Technologies]). GolgiStop and GolgiPlug (BD Biosciences) were used according to the manufacturer's instructions and added to the co-culture. Activation of the CD8+ T-cells was analyzed by flow cytometry. T-cells were transferred to a 96-well plate after co-culturing and surface staining of CD8 (BD Biosciences) was performed. Additionally, a viability dye-Cyan (EBioscience) was added. After fixation and permeabilization (BD Cytofix/Cytoperm Fixation/Permeabilization Kit), intracellular staining with FITC-coupled anti-IFNγ (BD Biosciences) and PE/Cy7-coupled anti-TNFα (BioLegend) was performed. Cells were measured on the FACS Canto (BD Biosciences). As positive controls, T-cells were cultured with phorbol 12-myristate 13-acetate (PMA) (2 µg/mL; Sigma-Aldrich) and ionomycin (40 µg/mL; Sigma-Aldrich) or with 15 µM HLA-B-specific CMV peptide for 5 hr.

## Statistics

Calculations based on flow cytometric results were performed from the median fluorescence intensity (MFI) after subtraction of background signal (secondary antibody or isotype control) and are shown with the standard error of the mean (SEM). For statistical analysis of the siRNA infection experiments and of the allotype-specific regulation of HLA-I surface expression after transient transfection, one-way ANOVA followed by Dunnett's multiple comparison test was performed as recommended by the GraphPad Prism software (v8). The correlation analysis (two-tailed) was performed with the same software. A p-value <0.05 was considered significant (*p<0.05; **p<0.005; ***p<0.0005). The radar chart was generated using Microsoft Excel (2016).

## Materials availability

Requests for resources should be directed to the corresponding author.

## Acknowledgements

We are thankful for the technical support by Magdalena Schwarzmüller and Lukas Göpfrich. The work was supported by the Deutsche Forschungsgemeinschaft through the TR 1208/1-1 (MT) and the FOR2830: DO 1275/7-2 (LD), ER 927/1-2 (FE), HA 6035/2-2 (AH), HE 2526/9-2 (HH), and SCHL 1888/8-2 (AS).

## Additional information

### Funding

| Funder | Grant reference number | Author |
|---|---|---|
| Deutsche Forschungsgemeinschaft | HA 6035/2-2 | Anne Halenius |
| Deutsche Forschungsgemeinschaft | TR 1208/1-1 | Mirko Trilling |
| Deutsche Forschungsgemeinschaft | ER 927/1-2 | Florian Erhard |
| Deutsche Forschungsgemeinschaft | DO 1275/7-2 | Lars Dölken |
| Deutsche Forschungsgemeinschaft | SCHL 1888/8-2 | Andreas Schlosser |
| Deutsche Forschungsgemeinschaft | HE 2526/9-2 | Hartmut Hengel |

The funders had no role in study design, data collection and interpretation, or the decision to submit the work for publication.

## Author contributions
Carolin Gerke, Conceptualization, Formal analysis, Investigation, Methodology, Writing – original draft; Liane Bauersfeld, Formal analysis, Methodology, Writing – review and editing; Ivo Schirmeister, Formal analysis, Validation, Investigation, Visualization, Writing – review and editing; Chiara Noemi-Marie Mireisz, Formal analysis, Investigation, Visualization, Writing – review and editing; Valerie Oberhardt, Methodology, Writing – review and editing; Lea Mery, Validation, Investigation, Visualization, Writing – review and editing; Di Wu, Investigation, Visualization; Christopher Sebastian Jürges, Formal analysis, Writing – review and editing; Robbert M Spaapen, Vu Thuy Khanh Le-Trilling, Mirko Trilling, Lars Dölken, Resources, Writing – review and editing; Claudio Mussolino, Wolfgang Paster, Resources, Methodology, Writing – review and editing; Florian Erhard, Formal analysis, Investigation, Visualization, Writing – original draft; Maike Hofmann, Frank Momburg, Resources, Validation, Methodology, Writing – review and editing; Andreas Schlosser, Supervision, Funding acquisition, Validation, Investigation, Methodology, Writing – review and editing; Hartmut Hengel, Resources, Funding acquisition, Validation, Writing – review and editing; Anne Halenius, Conceptualization, Supervision, Funding acquisition, Validation, Investigation, Visualization, Methodology, Writing – original draft

## Author ORCIDs
Carolin Gerke ⓘ https://orcid.org/0000-0001-6293-2741
Vu Thuy Khanh Le-Trilling ⓘ https://orcid.org/0000-0002-2733-3732
Mirko Trilling ⓘ https://orcid.org/0000-0003-3659-3541
Hartmut Hengel ⓘ https://orcid.org/0000-0002-3482-816X
Anne Halenius ⓘ https://orcid.org/0000-0001-6335-1017

## Ethics
Written informed consent was received from participants prior to inclusion in the study. The project has been approved by the ethical committee at the Albert-Ludwigs-University Freiburg (22-1196).

## Decision letter and Author response
Decision letter https://doi.org/10.7554/eLife.85560.sa1
Author response https://doi.org/10.7554/eLife.85560.sa2

# Additional files

## Supplementary files
MDAR checklist

## Data availability
Already published data used for our analysis can be found at these sites: dRNA-seq and STRIPE-seq data at NCBI Gene Expression Omnibus, GEO (accession number GSE191299), PRO(cap)-seq at GEO (GSE113394), and Ribo-seq at GEO (GSE41605), PacBio and MinION at the European Nucleotide Archive (accession number PRJEB25680), and protein expression data from the supplementary material of the reference (*Weekes et al., 2014*). These have been integrated into a genome browser and a web-based visualization platform (https://doi.org/10.5281/zenodo.5801030 and https://erhard-lab.de/web-platforms). PRICE is available at https://github.com/erhard-lab/price (*Erhard, 2018*) the gedi toolkit at https://github.com/erhard-lab/gedi (*Erhard, 2024*) and iTiSS at https://github.com/erhard-lab/iTiSS (*Jürges, 2021a*). The MetagenePlot, a module for gedi, is available at https://github.com/erhard-lab/MetagenePlot (*Jürges, 2021b*). The source code of the gedi toolkit is available at https://github.com/erhard-lab/gedi. The source code for additional custom scripts can be found at Zenodo (https://doi.org/10.5281/zenodo.5801030). The mass spectrometry proteomics data have been deposited to the ProteomeXchange Consortium via the PRIDE (*Perez-Riverol et al., 2022*) partner repository with the dataset identifier PXD051190.

The following dataset was generated:

| Author(s) | Year | Dataset title | Dataset URL | Database and Identifier |
|---|---|---|---|---|
| Gerke C, Bauersfeld L, Schirmeister I, Mireisz CNM, Oberhardt V, Mery L, Wu D, Jürges CS, Spaapen RM, Mussolino C | 2024 | Multimodal HLA-I genotype regulation by human cytomegalovirus US10 and resulting surface patterning | https://www.ebi.ac.uk/pride/archive/projects/PXD051190 | PRIDE, PXD051190 |

The following previously published datasets were used:

| Author(s) | Year | Dataset title | Dataset URL | Database and Identifier |
|---|---|---|---|---|
| Nilson KA, Parida M, Ball CB, Price DH, Li M, Meier JL | 2019 | Nucleotide resolution of RNA polymerase II transcription in human cytomegalovirus | https://www.ncbi.nlm.nih.gov/geo/query/acc.cgi?acc=GSE113394 | NCBI Gene Expression Omnibus, GSE113394 |
| Balázs Z, Tombácz D, Szűcs A, Snyder M, Boldogkői Z | 2018 | Dual-platform long-read RNA sequencing of the human cytomegalovirus lytic transcriptome | https://www.ebi.ac.uk/ena/browser/view/PRJEB25680 | European Nucleotide Archive, PRJEB25680 |
| Stern-Ginossar N, Weisburd B, Michalski A, Le VT, Hein MY, Huang SX, Ma M, Shen B, Qian SB, Hengel H, Mann M, Ingolia NT, Weissman JS | 2012 | Decoding human cytomegalovirus using ribosome profiling | https://www.ncbi.nlm.nih.gov/geo/query/acc.cgi?acc=GSE41605 | NCBI Gene Expression Omnibus, GSE41605 |
| Lodha M, Jürges CS | 2024 | Multi-omics reveals principles of gene regulation and pervasive non-productive transcription in the human cytomegalovirus genome | https://www.ncbi.nlm.nih.gov/geo/query/acc.cgi?acc=GSE191299 | NCBI Gene Expression Omnibus, GSE191299 |

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
