## [Editor Report]

HLA class I (HLA-I) molecules play a central role for Natural killer- as well as T-cell-mediated responses crucial to control the herpesvirus human cytomegalovirus (HCMV), and there are multiple viral immune evasins that target the protective function of HLA-I. This valuable study reports on the HCMV immune evasin US10 and shows that it recognizes and binds to all HLA-I (HLA-A,-B,-C,-E,-G) heavy chains and affects their function in different ways. These convincing findings reveal novel functions of the viral glycoprotein US10. The study improves our understanding of this complex and potentially life-threatening herpesvirus and its interaction with our immune system.

---

## [Decision Letter]

**Decision letter after peer review:**

Thank you for submitting your article "Multimodal HLA-I genotype regulation by human cytomegalovirus US10 and resulting surface patterning" for consideration by *eLife*. Your article has been reviewed by 3 peer reviewers, and the evaluation has been overseen by a Reviewing Editor and Satyajit Rath as the Senior Editor. The following individual involved in review of your submission has agreed to reveal their identity: Malgorzata A. Garstka (Reviewer #1).

Essential revisions (for the authors):

Overall, the three reviewers all agree that the results are interesting and valuable, and the experiments performed are biochemically very well done. However, they all also agree that a substantial amount of additional and more direct mechanistic studies of US10-mediated MHC class I down regulation would be needed to make this manuscript interesting for a broader audience.

They also all agree that the description of the transcription studies takes focus away from the role of US10 as a functional immunoevasin and should be pared down.

In addition, the reviewers pointed out the discussion about TAPBPR to be speculative, mainly because HLA-A*68, HLA-A*02:01 and A*24:02 were found to be TAPBPR binders, while HLA-A*03:01 and HLA-A*01:01 were not [Ilca et al., 2019, doi: 10.1016/j.celrep.2019.09.074]. In their discussion the reviewers pointed out that maybe differences in sequence/structures could explain the differential effect of US10 inhibition.

In addition, they stated that the model of US10-mediated retention of different HLA molecules is inconsistent with the binding studies of US10 with the HLA molecules and the PLC.

To get more mechanistic insights into US10's role for viral immune evasion, please address the following points of the reviewers:

1) US10 could potentially block the interaction with the PLC by, e.g., a) blocking the binding of beta2m (some indications in Figure 4: less W6/32-IPed B*44:02 in the presence of US10 than in the case of other alleles), b) blocking interaction with tapasin or c) calreticulin. As calreticulin binds to sugars, it is not allele-specific. Please address these possibilities experimentally.

2) US10 could bind beta2m before MHC class I could do so. Therefore, please analyse the interaction with beta2m (IP with BBM.1 antibody (-/+US10, IB: anti-HA, anti-US10), and/or whether US10 can bind before MHC class I binds beta2m [by a short pulse (60s) and short chase (1, 2, 5, 10, 20 min), resolve all bands: HA, US10, beta2m].

3) Binding of MHC class I with beta2m and tapasin should be studied using an overexpression system with a single MHC class I allele at the time, which may allow for differentiation between HLA-B and HLA-C alleles.

*Reviewer #1 (Recommendations for the authors):*

This is important work. I have a few specific points that may improve the manuscript.

Lines 129-132: "Interestingly, we observed that HLA-A*68:02 was less affected by this than HLA-B*15:03/C*12:03; the percentage of HLA-A*68:02 in the PLC as compared to the total amount of HLA-I in the PLC was increased in US10-expressing cells (Figure 2D). Hence, HLA- A*68:02 access to the PLC is less disturbed by US10 than observed for HLA-B*15:03/-C*12:03."

Figure 2C presents the interaction of HLA-A*68:01, B*15:03, and C*12:03 with ERp57 or tapasin in the presence or the absence of US10. The authors claim that although US10 inhibits interaction with PLC for all three alleles, HLA-B*68:02 is less affected. The quantification in D may be misleading as it demonstrates only data for HLA-B*68:02. I suggest representing the ratio of HLA-B*68:02/ERp57 (or tapasin) and B*15:03+C*12:03/ERp57 (or tapasin) that may allow more direct comparison between A and B/C allotypes in the presence and absence of US10 in HeLa cell line.

Suppl. Figure 1. Lines 136-138 "Furthermore, we demonstrate here that under the same experimental conditions, US2 and US3 were co-immunoprecipitated by anti-ERp57, whereas this was not the case for US10 (Suppl. Figure 1)." The authors should include entire membrane and indicate ERp57 band. Besides, the included exposure allows seeing US2HA co-IP with ERp57 but not US3. Could it be that US10 which is of lower molecular weight is below the detection limit? Including longer exposure may strengthen the authors' argument.

Lines 174-178 "Interestingly, in samples where US10 was strongly co-immunoprecipitated with the mAb W6/32 also induced HC/β2m assembly was observed, particularly pronounced for HLA-C*05:01, -C*07:02, and – G*01:01. In contrast, the tapasin-dependent HLA-B*44:02 allotype showed a reduced level of β2m assembly in the presence of US10 (Figure 4D-E). " It seems that US10 expression enhanced MHC class I (either IP with W6/32 or HA antibody). To show that there is an additional effect on b2m-bound HLA-C*05:01, -C*07:02, -G*01:01, and B*44:02, I suggest including a graph with the comparison of these two effects, specifically, a ratio of US10/control for MHC class I precipitated with W6/32 next to the ratio of US10/control for MHC class I precipitated with HA.

Lines 320-321 "Our data support a model in which the affinity of US10 for HLA-A allotypes is lower than for HLA-B." Figure 1C shows similar binding of US10 to all studied MHC class I alleles (all forms). Figure 4A shows similar binding of US10 to HLA-A*02:01 and B*07:02 (total or b2m-bound form), thus the affinity of US10 to HLA-A and HLA-B alleles seem similar.

Lines 387-388: "US10 is able to bind to all HLA-I molecules early after their synthesis and prior to dimerization with β2m (Figure 8)." – The experiments may not support the above claim. In Figure 1C, 2A, 2C, both US10, and b2m are present in co-IP samples. In Figure 4A, the membrane is cut above b2m so it is not possible to access whether b2m is in the complex with MHC class I and US10. Moreover, a low signal of B*44:02 in Figure 4A may hinder the assessment of whether it binds to US10 while being in the complex with b2m. Short metabolic labeling would be needed to assess whether US10 binds to MHC class I before the MHC-I-b2m complex is formed.

*Reviewer #2 (Recommendations for the authors):*

These are proposed points that would provide significance to the findings:

1) US10 has previously been described as an immune-evasin molecule with specificity for HLA-I alleles and a different mechanism. The differences in the US10 mechanism of action has not been demonstrated with experiments.

2) Figure 4 illustrates binding assays using W6/32 and an ant-HA antibody. The class I/US10 interaction does not correlate with class I down regulation shown in Figure 1. Thus, the mechanism of US10 recognition of MHC class I that contributes to its surface expression is not consistent with the model of downregulation. These findings should be clarified.

3) The analysis of US10 impacting HLA-down regulation during a virus infection is difficult to evaluate due to the small differences in MFI on flow cytometry and that using US10 siRNA impacts US11 expression. Thus, it is difficult for the authors to make bold conclusions for Figure 7. In addition, data supporting the US10 or US11 levels during infection would provide more confidence in the studies.

*Reviewer #3 (Recommendations for the authors):*

The cell surface staining experiments of Figures 1 and 2 are solid. Although the data on C*05:01 and C*07:02 show a difference in these two allelomorphs, in some places in the text there seems to be a pooling of HLA-C and -G rather than to make the distinction in some alleles. Although the 2D plot in Figure 3B seems to illustrate the relationship of the Tapasin independence, there are no error bars or statistics with panel A. This needs better explanation. Also, K(b) is indeed tapasin dependent in the mouse-that needs better explanation. It would have been nice to continue the analysis of B*4402 vs B*44:05, the classical tapasin dependent/independent pair that differ by a single residue polymorphism. The digression into the transcriptional regulation of US10 and US11 is rather confusing when the main point of the papeer seems to be the effect of the translated US10 on PLC dependent peptide editing. Supplemental Figure B is labelled "in Tapasin dependent order", but indeed the order is tapasin independent. The only functional experiment addressing US10 as a potential immunoevasin is shown in Suppl. Figure 6B. This should be explained more completely and used to justify the important biological potential. Overall, the observations are sound, with some concern as to the full statistical significance of some of the comparisons. I would certainly urge the authors to pare down the digression to the transcription, as this seems to wander from the main point of the paper.

---

## [Author Response]

Essential revisions (for the authors):To get more mechanistic insights into US10's role for viral immune evasion, please address the following points of the reviewers:1) US10 could potentially block the interaction with the PLC by, e.g., a) blocking the binding of beta2m (some indications in Figure 4: less W6/32-IPed B*44:02 in the presence of US10 than in the case of other alleles), b) blocking interaction with tapasin or c) calreticulin. As calreticulin binds to sugars, it is not allele-specific. Please address these possibilities experimentally.

We have included new data showing that US10 can bind to assembled HLA-A*02:01 and HLA-B*07:02 if they are retained in the ER by brefeldin A (Figure 4F and Figure 3—figure supplement 1D). This demonstrates that US10 and β_2_m binding to the HC is not mutually exclusive.

Upon re-introduction of β_2_m into β_2_m-KO cells, US10 interaction with HLA-A*02:01 and HLA-B*07:02 was reduced, but not with HLA-C*05:01 (Figure 4D). This shows that the affinity of US10 for assembled HLA-A and -B is lower than for HLA-G and some -C, indicating a difference in the quality of the binding, which results in strong retention of HLA-G and some -C, but not of -A and -B.

The new quantitative HLA-I ligandome analysis support the idea that US10 affects HLA-I genotypes in different manner: the quality control of HLA-B*15:03 was strongly reduced, leading to loss of HLA-B*15:03 expression, while peptide-loaded HLA-C*12:03 was stabilized in the ER. No effects were observed on HLA-A*68:02 ligandome.

Furthermore, we found that US10 may interact with tapasin directly in an HLA-I-independent manner (Figure 2E, see also reply to point 3). This is different from our previous submission, in which we concluded that US10 is not interacting with the PLC. However, we now labeled the cells for a longer time and transiently transfected tapasin to induce a strong expression. This indeed allowed the detection of an interaction between US10 and tapasin. We therefore hypothesize that tapasin will be involved in functional aspects of US10. This will be investigated in more detail in a separate study.

2) US10 could bind beta2m before MHC class I could do so. Therefore, please analyse the interaction with beta2m (IP with BBM.1 antibody (-/+US10, IB: anti-HA, anti-US10), and/or whether US10 can bind before MHC class I binds beta2m [by a short pulse (60s) and short chase (1, 2, 5, 10, 20 min), resolve all bands: HA, US10, beta2m].

This is an interesting experimental setup. We performed a pilot experiment, but quite as we expected, it was not possible to visualize US10 in complex with MHC-I after a short labelling. We have previously observed that accumulation of newly synthesized US10 in a complex with MHC-I is slow (for example see Figure 2A). We believe that the reason for this is a slow insertion of newly synthesized US10 into the ER membrane.

We did not find direct binding of US10 to β_2_m (Figure 4—figure supplement 1A, lane 16). Therefore, US10 binding to assembled HLA-I is conferred through the HC.

We generated HLA-I and β_2_m double knock-out HeLa cells. In these cells US10 binding to HA-tagged HLA-A*02:01, -B*07:02, -B*44:02, and -C*05:01 was investigated. Using the anti-HA antibody, US10 was co-immunoprecipitated with all four HLA-I molecules (Figure 4D), demonstrating that US10 can bind to the free HCs of these HLA-I. β_2_m overexpression reduced US10 binding to HLA-A*02:01, -B*07:02, and -B*44:02. While this is consistent with the idea that β_2_m could displace US10 from the HLA-I HC, it could also mirror the level of transport of HLA-I out of the ER after expression of β_2_m. Indeed, upon treatment of cells with brefeldin A, which blocks trafficking between the ER and the Golgi compartment, induced interaction of US10 with HLA-A*02:01 and HLA-B*07:02 was observed (Figure 4F and Figure 4—figure supplement 1E). Therefore, it is possible to force this interaction, demonstrating that β_2_m and US10 binding to the HLA-I HC is not mutually exclusive.

3) Binding of MHC class I with beta2m and tapasin should be studied using an overexpression system with a single MHC class I allele at the time, which may allow for differentiation between HLA-B and HLA-C alleles.

Regarding β_2_m, please see our above comment.

During revision, we discovered an interaction between US10 and tapasin (Figure 2D), which contrasts with our initial findings. The actual reason for this re-investigation was the result of an interactome analysis that we have performed with US10 (immunoprecipitation and subsequent LC-MS/MS analysis of interaction partners). In addition to HLA-I allomorphs, in this interactome we detected pronounced interactions with all proteins of the PLC. It was therefore necessary to reassess the submitted data. Insufficient labeling during previous experiments is one reason for the obscured interaction, suggesting that US10 interaction with the PLC is not a dynamic event, i.e. that formed complexes are not in frequent exchange with newly synthesized proteins. Even under prolonged labeling times, the interaction with the PLC was difficult to visualize. However, upon tapasin overexpression a convincing detection of the US10-tapasin interaction was possible; the interaction was observed using both anti-ERp57 and anti-US10 antibodies. This finding raises intriguing questions that warrant further investigation. However, due to the number of experiments that should be performed, we believe it is more suitable to investigate these questions in a subsequent study. Therefore, we have not included further data on the association of the PLC with US10 and HLA-I in this manuscript.

However, the new HLA-I ligandome analysis that we provided strongly supports our conclusions regarding the genotype-dependent effects of US10. It shows that HLA-B*15:03 was destabilized by US10 and that quality control of peptide binding was missing. On the contrary, US10 stabilized peptide-loaded HLA-C*12:03 in the ER (Figure 5). No change of the HLA-A*68:02 ligandome was observed

4) Please also address the specific concerns and suggestions of the three reviewers listed below.This study presents a useful finding on a virally encoded immune-evasin which differentially inhibits antigen presentation by cellular protein complexes called Major histocompatibility complex (MHC) class I, thereby diminishing the activation of cytotoxic T cells. The evidence supporting the claims of the authors is solid, although the addition of more mechanistic insights would strengthen the study. The work will be of interest to virologists and immunologists working on the adaptive immune response to herpesviral infection. Some conclusions would require additional experimental support.Reviewer #1 (Public Review):HMCV encodes various immunoevasins to inhibit being presented by MHC class I molecules to the cytotoxic cells of the immune system. Here, the authors studied the role and specificity of US10, a relatively uncharacterized immunoevasin from HCMV. They found that US10 differentially affects antigen presentation by different MHC class I allotypes. HLA-A and certain HLA-B and C alleles (so-called "tapasin-independent") were unaffected, while other HLA-B and C alleles (so-called "tapasin-dependent") as well as HLA-G were negatively affected. US10 can bind to different MHC class I allotypes, which inhibits their incorporation into peptide loading complex and slowers maturation. By comparing US10 to the other well-studied immunoevasins from HCMV, US2, US3, and US11, the authors demonstrated only partial overlap between them suggesting the cumulative action of immunoevasins in inhibiting MHC class I antigen presentation of HMCV epitopes. This work contributes to our understanding of the complex immune evasion mechanism by HCMV.The strengths include using a broad use of available techniques, including overexpression of US10 and US10 siRNA in the infection context that allowed comparison of its net and cumulative effects. Bioinformatic analysis of US10 and US11 to describe how transcription and expression of these two gene products contribute to the control of immunoevasion by HCMV. The conclusions are mostly supported by the experiments.Reviewer #2 (Public Review):The manuscript entitled " Multimodal HLA-I genotypes regulation by human cytomegalovirus US10 and resulting surface patterning" by Gerke et al describes the biochemical analysis of US10-mediated down regulation of HLA-I molecules. The authors systemically examine the surface expression of different HLA-I alleles in cells expressing US10 and interactions of US10 with HLA-I and antigen presentation machinery. Further, studies examined genotypic and allotypic differences during expression of US10/US11 transcripts suggest a different allelic class I downregulation. In general, the authors have included data supporting the major claims. Yet, the conclusions and findings of the study only marginally advance the overall understanding of HCMV viral evasion and the mechanism of US10 function.Strengths:The studies are well characterized and the studies utilize diverse HLA-I and HCMV viral molecules. The biochemistry is excellent and is of high quality. Importantly, the study describes HLA-I allelic specific HCMV down regulation at the cell surface and molecular levels.Weaknesses:1) The authors use over expressive language such as "strong binding" that does not have a quantitative value and it is relative to the specific assay with only small differences among the factors.

We have changed the language to avoid non-quantitative expressions.

2) The US10 binding to the HLA-I did not correlate with class I surface levels suggesting that binding to the APC machinery (Figure 1); hence, why does the binding of US10 to the APC define its mechanism of action.

We hypothesized that since binding to HLA-I allomorphs did not correlate with surface expression, further factors could be involved in regulation. Since the PLC (APC machinery) plays a major role for HLA-I expression, it was relevant to investigate this. The new data underlines the importance of the PLC for US10-mediated HLA-I regulation.

3) The innovative and significant aspects of the study are limited. The study does not delineate the US10 mechanism of action or show data in which US10-mediated MHC class I down regulation impacts adaptive or innate immune function.

These remarks are important. We want to emphasize the variable impact of US10 on HLA-I. To our knowledge previous studies have not uncovered genotype-dependent effects on HLA-I as distinct as those observed with US10, indicating that US10 may exploit aspects of HLA-I that are yet to be fully elucidated. Therefore, confirming these findings is crucial for our study. The quantitative analysis of the HeLa HLA-I ligandome in US10-expressing cells strongly supports this conclusion. The precise quantification of HLA-I peptide ligands was made possible through collaboration with Dr. Andreas Schlosser from Würzburg, Germany, who possesses profound expertise in this specific method. Thus, in our opinion, this revision has enabled us to advance innovation and, importantly, enhance the significance of our study.

Reviewer #3 (Public Review):Correlation of the HLA-B effects with previously demonstrated allelic differences in dependence on the peptide loading complex (PLC) component chaperone/editor tapasin and demonstration that US10 does not bind the PLC reflect on possible mechanisms of US10 function. Thus, this paper adds new information that may be integrated into evolving models of the steps of MHC-I dependent antigen presentation and how viruses counter immune recognition for their own benefit. Clearer focus on the proposed models for the function of US10 and its mechanism--i.e. what experiments address the mechanism and what additional finding might clarify the mechanism would be helpful.